# Self-ensemble Adversarial Training for Improved Robustness

**Hongjun Wang[1]**    **Yisen Wang[1,2][†]**
[1] Key Lab. of Machine Perception (MoE), School of Artificial Intelligence, Peking University
[2] Institute for Artificial Intelligence, Peking University

## Abstract

Due to numerous breakthroughs in real-world applications brought by machine intelligence, deep neural networks (DNNs) are widely employed in critical applications. However, predictions of DNNs are easily manipulated with imperceptible adversarial perturbations, which impedes the further deployment of DNNs and may result in profound security and privacy implications. By incorporating adversarial samples into the training data pool, adversarial training is the strongest principled strategy against various adversarial attacks among all sorts of defense methods. Recent works mainly focus on developing new loss functions or regularizers, attempting to find the unique optimal point in the weight space. But none of them taps the potentials of classifiers obtained from standard adversarial training, especially states on the searching trajectory of training. In this work, we are dedicated to the weight states of models through the training process and devise a simple but powerful *Self-Ensemble Adversarial Training* (SEAT) method for yielding a robust classifier by averaging weights of history models. This considerably improves the robustness of the target model against several well known adversarial attacks, even merely utilizing the naive cross-entropy loss to supervise. We also discuss the relationship between the ensemble of predictions from different adversarially trained models and the prediction of weight-ensembled models, as well as provide theoretical and empirical evidence that the proposed self-ensemble method provides a smoother loss landscape and better robustness than both individual models and the ensemble of predictions from different classifiers. We further analyze a subtle but fatal issue in the general settings for the self-ensemble model, which causes the deterioration of the weight-ensembled method in the late phases[*].

## 1 Introduction

Deep learning techniques have showed promise in disciplines such as computer vision (Krizhevsky et al., 2012; He et al., 2016), natural language processing (Vaswani et al., 2017; Devlin et al., 2019), speech recognition (Sak et al., 2015; Wang et al., 2017) and even in protein structural modeling (Si & Yan, 2021; Jumper et al., 2021). However, even for those efforts surpassing human-level performance in several fields, deep learning based methods are vulnerable to adversarial examples generated by adding small perturbations to natural examples (Szegedy et al., 2014; Goodfellow et al., 2015).

Following the discovery of this adversarial vulnerability (Ma et al., 2020; Wang et al., 2020b; Niu et al., 2021), numerous defense approaches for protecting DNNs from adversarial attacks have been proposed (Papernot et al., 2016; Xu et al., 2018; Liao et al., 2018; Ross & Doshi-Velez, 2018; Bai et al., 2019; Qin et al., 2020; Wang et al., 2020a; Bai et al., 2021; Huang et al., 2021). Because it is a straightforward and effective data-driven strategy, among these methods, adversarial training (Madry et al., 2018) has become the de-facto way for strengthening the robustness of neural networks. The core of adversarial training is proactively inducing a robust model by forcing DNNs to learn adversarial examples generated from the current model, which helps DNNs gradually make up for their deficiencies. Therefore, it is a practical learning framework to alleviate the impact of adversarial attacks.

As a saying goes, a person may go faster but a team can go farther. An ensemble of classifiers fused by output aggregates the collective wisdom of all participated classifiers, which outperform the

---

[†]Correspondence to: Yisen Wang (yisen.wang@pku.edu.cn)

[*]Code is available at `https://github.com/whj363636/Self-Ensemble-Adversarial-Training`

decision made by a single classifier. Many ensemble methods (Tramèr et al., 2018; Pang et al., 2019) have shown their abilities to boost adversarial robustness. Despite averaged predictions of different models enables the aggregation against a large number of attacks, it is still a heavy computing burden to adversarially train all the member models. What is worse, the architectures and parameters of *every* member model, as well as their related contribution factors, should be safely saved and need performing forward propagation for every member repeatedly, which is a nightmare when dealing with massive components. Despite the fact that the ensemble technique adopts a variety of architectures and parameters, these models may have similar decision-making boundaries. Therefore, it is natural to wonder:

> *Could we develop the potential of the classifier itself to build a robust model rather than rely on multiple seemingly disparate classifiers? Could we make use of states on the searching trajectory of optimization when performing adversarial training?*

Motivated by these questions, we discuss the relationship between an ensemble of different classifiers fused by predictions and by parameters, and present a computation-friendly self-ensemble method named SEAT. In summary, this paper has the following contributions:

- By rethinking the standard adversarial training and its variants, instead of developing a well-designed loss to find a single optimal in the weight space, we design an efficient algorithm called *Self-Ensemble Adversarial Training* (SEAT) by uniting states of every history model on the optimization trajectory through the process of adversarial training. Unlike the traditional ensemble technique, which is extremely time-consuming and repeating adversarial training procedures for individuals, SEAT simply requires training once.

- We present a theoretical explanation of the differences between the two ensemble approaches discussed above and visualize the change of loss landscape caused by SEAT. Furthermore, while using data augmentation of external data can experimentally fix the deterioration during the late stages (Gowal et al., 2020; Rebuffi et al., 2021), we further investigate a subtle but fatal intrinsic issue in the general settings for the self-ensemble model, which causes the deterioration of the weight-ensembled method.

- We thoroughly compare adversarial robustness of SEAT with other models trained by the state-of-the-art techniques against several attack methods on CIFAR-10 and CIFAR-100 datasets. Results have shown that the SEAT method itself can efficiently search in the weight space and significantly improve adversarial robustness.

## 2 REVIEW: STANDARD ADVERSARIAL TRAINING AND ITS SUCCESSORS

In this section, we first introduce basic concepts of adversarial training. Then we briefly review some variants and enhancements of standard adversarial training.

### 2.1 PRELIMINARIES

Consider a standard image task which involves classifying an input image into one of $C$ classes, let $x \in [0,1]^n$ denote the input natural image and $y \in \{1, 2, \ldots, C\}$ be the ground truth label, sampled from a distribution $\mathbb{D}$. For adversarial attack, the goal of the adversary is to seek a malignant example in the vicinity of $x$, by adding a human imperceptible perturbation $\varepsilon \in \mathbb{R}^n$. To exactly describe the meaning of "imperceptible", this neighbor region $\mathbb{B}_\varepsilon(x)$ anchored at $x$ with apothem $\varepsilon$ can be defined as $\mathbb{B}_\varepsilon(x) = \{(x', y) \in \mathcal{D} \mid \|x - x'\| \leq \varepsilon\}$. We also define a loss function: $\ell \overset{def}{=} \Theta \times \mathbb{D} \to [0, \infty)$, which is usually positive, bounded, and upper-semi continuous (Blanchet & Murthy, 2019; Villani, 2003; Bartlett & Mendelson, 2001) for all $\boldsymbol{\theta} \in \Theta$. Instead of the original formulation of adversarial training (Szegedy et al., 2014; Goodfellow et al., 2015), which mixes natural examples $x$ and adversarial examples $x'$ together to train models, the adversarial risk minimization problem can be formulated as a two-player game that writes as below:

$$\underbrace{\inf_{\boldsymbol{\theta} \in \Theta} \mathbb{E}_{(x,y) \sim \mathbb{D}}}_{\text{Outer minimization}} \underbrace{\sup_{x' \in \mathbb{B}_\varepsilon(x)} \ell\left(\boldsymbol{\theta}; x', y\right)}_{\text{Inner maximization}}. \tag{1}$$

In general, the adversary player computes more and more harmful perturbations step by step, where the PGD method (Madry et al., 2018) is commonly applied:

$$x^{t+1} = \Pi_{\mathbb{B}_\varepsilon(x)}\left(x^t + \kappa \operatorname{sign}\left(\nabla_x \ell(x, y)\right)\right), \tag{2}$$

where $\kappa$ is the step size and the generated samples at each iteration will be projected back to the $\mathbb{B}_\varepsilon(x)$. When finding $x'$ at their current optimal in the inner maximization, the model parameters will be updated according to these fixed samples only in the outer minimization.

## 2.2 RELATED WORK

Emerging adversarial training methods are primarily concerned with strengthening their robustness through empirical defense. Here we roughly separate these improvements into two categories: (1) *sample-oriented* (i.e. for $x'$ in the inner maximization) and (2) *loss-oriented* (i.e. for $\ell$ in the outer minimization) improvement.

From the side of the former, Baluja & Fischer (2017) utilize a GAN framework to create adversarial examples via the generative model. Wang et al. (2019) proposes a criterion to control the strength of the generated adversarial examples while FAT (Zhang et al., 2020) control the length of searching trajectories of PGD through the lens of geometry.

For the latter, a stream of works have been carried out with the goal of establishing a new supervised loss for better optimization. Usually, the loss $\ell$ in the both inner maximization and outer minimization[†] are cross-entropy defined as: $\ell_{ce} = -\boldsymbol{y}^T \log \boldsymbol{q}$ when given the probability vector $\boldsymbol{q} = [e^{f_{\boldsymbol{\theta}}(x')_1}, \cdots, e^{f_{\boldsymbol{\theta}}(x')_c}]/\sum_{k=1}^{c} e^{f_{\boldsymbol{\theta}}(x')_k}$. The addition of a regularizer to the vanilla loss is one branch of the modification. Representative works like ALP (Kannan et al., 2018), TRADES (Zhang et al., 2019), and VAT (Miyato et al., 2019) introduce a regularization term to smooth the gap between the probability output of natural examples and adversarial ones, which can be denoted as:

$$\ell_{reg} = \ell_{ce} + \eta \sum_{(x_i, y_i) \sim \mathcal{B}} \mathcal{R}(p(\boldsymbol{\theta}; x'_i, y_i), p(\boldsymbol{\theta}; x_i, y_i)), \tag{3}$$

where $\eta$ is the regularization hyper-parameter and $\mathcal{R}(\cdot, \cdot)$ is $l_2$ distance for ALP and VAT[‡], and Kullback-Leibler divergence for TRADES. $\mathcal{B}$ stands for the sampled minibatch. Another branch of the loss modification is to reweight the contributions of each instance in the minibatch based on their intrinsic characteristics. This can be formulated by:

$$\ell_{rew} = \sum_{(x_i, y_i) \sim \mathcal{B}} w(\boldsymbol{\theta}; x_i, x'_i, y_i)\ell_{ce}(\boldsymbol{\theta}; x_i, x'_i, y_i), \tag{4}$$

where $w(\cdot)$ is a real-valued function mapping the input instances to an importance score. MMA (Ding et al., 2020) separately considers the correct and incorrect classification on natural cases and switches between them by learning a hard importance weight:

$$\ell_{\text{MMA}} = \sum_{(x_i, y_i) \sim \mathcal{B}} w_1(\boldsymbol{\theta}; x_i, y_i)\ell_{ce}(\boldsymbol{\theta}; x_i, y_i) + w_2(\boldsymbol{\theta}; x_i, y_i)\ell_{ce}(\boldsymbol{\theta}; x'_i, y_i), \tag{5}$$

where $w_1$ and $w_2$ are the indicator function $\mathbb{1}(f_{\boldsymbol{\theta}}(x_i) \neq y_i)$ and $\mathbb{1}(f_{\boldsymbol{\theta}}(x_i) = y_i)$, respectively. Inspired by the geometry concern of FAT, GAIRAT (Zhang et al., 2021) evaluates $w(\boldsymbol{\theta}; x, y) = (1 + \tanh(c_1 - 10c_2 + 5))/2$ to adaptively control the contribution of different instances by their corresponding geometric distance of PGD, where $c_1$ is a hyper-parameter and $c_2$ is the ratio of the minimum successful iteration numbers of PGD to the maximum iteration step. Embracing both two schools of improvement, MART (Wang et al., 2020c) adds a new regularization term apart from KL divergence and explicitly assigns weights within regularizers:

$$\ell_{\text{MART}} = \ell_{ce} + \sum_{(x_i, y_i) \sim \mathcal{B}} w(\boldsymbol{\theta}; x_i, y_i)\mathcal{R}_{KL}(p(\boldsymbol{\theta}; x'_i, y_i), p(\boldsymbol{\theta}; x_i, y_i)) + \mathcal{R}_{mag}(p(\boldsymbol{\theta}; x'_i, y_i)), \tag{6}$$

where $\mathcal{R}_{KL}$ is Kullback-Leibler divergence same as TRADES and $w_i = (1 - p(\boldsymbol{\theta}; x_i, y_i))$ is a softer scheme when compared with MMA. The margin term $\mathcal{R}_{mag} = -\log(1 - \max_{k \neq y_i} p_k(\boldsymbol{\theta}; x'_i, y_i))$ aims at improving the decision boundary.

The above methods endeavor to solve the highly non-convex and non-concave optimization in Eqn 1. However, they assume that the optimal weights of a classifier appear in its prime and abandon all the states along the route of optimization, which are beneficial to approach the optimal.

---

[†]For simplification, here we ignore the implemented difference of loss in the inner maximization and the outer minimization (e.g. TRADES and MMA). The reference of $\ell$ can be contextually inferred easily.

[‡]For VAT, $\theta$ in the first term and the second term are different.

---

**Algorithm 1** Self-Ensemble Adversarial Training (SEAT)

---

**Input:** A DNN classifier $f_{\boldsymbol{\theta}}(\cdot)$ with initial learnable parameters $\boldsymbol{\theta}_0$ and loss function $\ell$; data distribution $\mathbb{D}$; number of iterations $N$; number of adversarial attack steps $K$; magnitude of perturbation $\varepsilon$; step size $\kappa$; learning rate $\tau$; exponential decay rates for ensembling $\alpha$; constant factor $c$.

Initialize $\boldsymbol{\theta} \leftarrow \boldsymbol{\theta}_0, \tilde{\boldsymbol{\theta}} \leftarrow \boldsymbol{\theta}$.

**for** t $\leftarrow 1, 2, \cdots, N$ **do**

    Sample a minibatch (x, y) from data distribution $\mathbb{D}$

    $x'_0 \leftarrow x + \varepsilon, \varepsilon \sim \text{Uniform}(-\varepsilon, \varepsilon)$.

    **for** k $\leftarrow 1, 2, \cdots, K$ **do**

        $x'_k \leftarrow \Pi_{x'_k \in \mathbb{B}_\varepsilon(x)} \left( \kappa \operatorname{sign} \left( x'_{k-1} + \nabla_{x'_{k-1}} \ell(\boldsymbol{\theta}; (x'_k, y)) \right) \right)$

    **end for**

    $\boldsymbol{g}_{\boldsymbol{\theta}_t} \leftarrow \mathbb{E}_{(x,y)} [\nabla_{\boldsymbol{\theta}_t} \ell(\boldsymbol{\theta}_t; (x'_k, y))]$ in every minibatch

    Calculate $\tau_t$ according to the current iterations

    $\boldsymbol{\theta}_t \leftarrow \boldsymbol{\theta}_t - \tau_t \boldsymbol{g}_{\boldsymbol{\theta}_t}$

    $\alpha' \leftarrow \min \left( \alpha, \frac{t}{t+c} \right)$

    $\tilde{\boldsymbol{\theta}} \leftarrow \alpha' \tilde{\boldsymbol{\theta}} + (1 - \alpha') \boldsymbol{\theta}_t$

**end for**

**Return** A self-ensemble adversarial training model $f_{\tilde{\boldsymbol{\theta}}}(\cdot)$

---

## 3 METHODOLOGY

In this section, we first discuss the traditional ensemble methods based on predictions of several individual classifiers. Then, we propose a Self-ensemble Adversarial Training (SEAT) strategy fusing weights of individual models during different periods, which is both intrinsically and computationally convenient. Following that, we further provide theoretical and empirical analysis on why the prediction of such an algorithm can make a great difference from that of simple ensembling classifiers.

### 3.1 PREDICTION-ORIENTED ENSEMBLE

Ensemble methods are usually effective to enhance the performance (Caruana et al., 2004; Garipov et al., 2018) and improve the robustness (Tramèr et al., 2018; Pang et al., 2019). For clarity, we denote $\mathcal{F}$ as a pool of candidate models where $\mathcal{F} = \{ f_{\boldsymbol{\theta}_1}, \cdots, f_{\boldsymbol{\theta}_n} \}$. To represent the scalar output of the averaged prediction of candidate classifiers over $C$ categories, we define the averaged prediction $\bar{f}$ involving all candidates in the pool $\mathcal{F}$ as:

$$
\bar{f}_{\mathcal{F}}(x, y) = \sum_{i=1}^{n} \beta_i f_{\boldsymbol{\theta}_i}(x, y)
$$

$$
s.t. \sum_{i=1}^{n} \beta_i = 1, \tag{7}
$$

where $\beta$ represents the normalized contribution score allotted to the candidates. It is logical to assume that $\forall \beta_i > 0$ since each candidate model must be well-trained, otherwise it will be expelled from the list. Note that here we only discuss whether the participated model performs well on its own, not whether it has a positive impact on the collective decision making, which is a belated action.

### 3.2 SELF-ENSEMBLE ADVERSARIAL TRAINING

In order to average weights of a model, it is natural to choose an analogous form as Eqn 7 to obtain the predicted weights: $\tilde{\boldsymbol{\theta}} = \sum_{t=1}^{T} \beta_t \boldsymbol{\theta}_t, s.t. \sum_{t=1}^{T} \beta_t = 1$. However, such a simple moving average cannot keenly capture the latest change, lagging behind the latest states by half the sample width. To address this problem, we calculate the optimal weights by characterizing the trajectory of the weights state as an exponential moving average (EMA) for measuring trend directions over a period of time. Intuitively, EMA not only utilizes recent proposals from the current SGD optimization but also maintains some influence from previous weight states:

$$
\tilde{\boldsymbol{\theta}}_T = \alpha \tilde{\boldsymbol{\theta}}_{T-1} + (1 - \alpha) \boldsymbol{\theta}_T
$$

$$
= \sum_{t=1}^{T} (1 - \alpha)^{1 - \delta(t-1)} \alpha^{T-t} \boldsymbol{\theta}_t, \tag{8}
$$

where $\delta(\cdot)$ is the unit impulse function (i.e., $\delta(0) = 1$, otherwise it is 0). Algorithm1 summarizes the full algorithm.

In consideration of both using the moving average technique, we wonder whether the averaged prediction has a link to the prediction of a weight-averaged model. In fact, they are indeed related to each other to some extent:

**Proposition 1.** *(Proof in Appendix B) Let $f_{\boldsymbol{\theta}}(\cdot)$ denote the predictions of a neural network parametrized by weights $\boldsymbol{\theta}$. Assuming that $\forall \boldsymbol{\theta} \in \Theta$, $f_{\boldsymbol{\theta}}(\cdot)$ is continuous and $\forall (x, y) \in \mathbb{D}$, $f_{\boldsymbol{\theta}}(x, y)$ is at least twice differentiable. Consider two points $\boldsymbol{\theta}_t, \tilde{\boldsymbol{\theta}} \in \Theta$ in the weight space and let $\boldsymbol{\xi} = \boldsymbol{\theta}_t - \tilde{\boldsymbol{\theta}}$, for $t \in \{1, 2, \cdots, T\}$, the difference between $\bar{f}_{\mathcal{F}}(x, y)$ and $f_{\tilde{\boldsymbol{\theta}}}(x, y)$ is of the second order of smallness if and only if $\sum_{t=1}^{T}(\beta_t \boldsymbol{\xi}^{\top}) = \mathbf{0}$.*

Based on Proposition 1, we could immediately obtain the below conclusions:

**Remark 1.** *Note that it always constructs ensembles of the well-matched enhanced networks to obtain stronger defense, so it evenly assigns $\beta_1 = \beta_2 = \cdots = \beta_n = 1/n$. However, things change in the self-ensemble setting since models obtained at relatively late stages will be much robust than the beginning. Based on this assumption, it has a non-decreasing sequence $\beta_1 \le \beta_2 \le \cdots \le \beta_n, s.t. \sum_{i=1}^{n} \beta_i = 1$. The inequality is tight only when the initial weight reaches its fixed point.*

In this case, the averaged prediction will predispose to the models obtained at the end of optimization and such a predisposition loses the superiority of ensemble due to the phenomenon of homogenization in the late phases. To provide an empirical evidence of homogenization of models at latter periods, we visualize the effect of homogenization along the training process in Figure 1a. We define the homogenization of a model as: $\triangle_e = \frac{1}{\|\mathcal{D}\|} \sum \min_{i \in [1,m]} |f_{\boldsymbol{\theta}_e}(x, y) - f_{\boldsymbol{\theta}_{e-i}}(x, y)|$, used for calculating the difference of the output of history models over a period of time $m$. Note that $\triangle_e$ becomes smaller and smaller along with epochs passing, which proves the existence of homogenization.

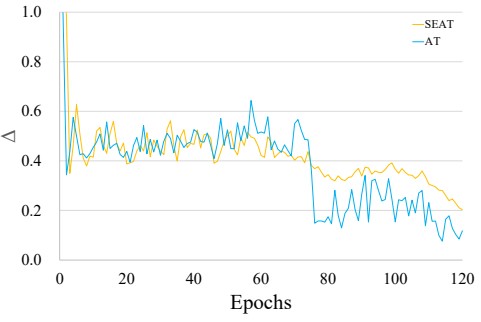
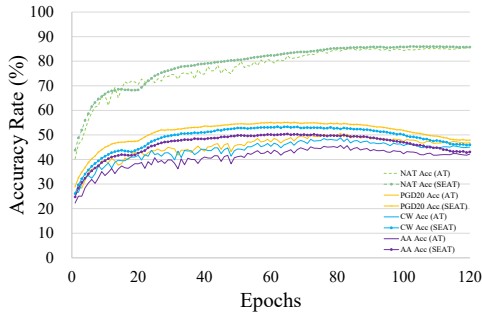

(a) The effect of homogenization of models.

(b) Robust accuracy of parameter-averaged ResNet18 against various attacks on CIFAR-10.

Figure 1: The overall analyse of AT and SEAT.

Here we formally state the gap between such an averaged prediction and the prediction given by SEAT in the following theorem:

**Theorem 1.** *(Proof in Appendix B) Assuming that for $i, j \in \{1, \cdots, T\}$, $\boldsymbol{\theta}_i = \boldsymbol{\theta}_j$ if and only if $i = j$. The difference between the averaged prediction of multiple networks and the prediction of SEAT is of the second order of smallness if and only if $\beta_i = (1 - \alpha)^{1 - \delta(i-1)} \alpha^{T-i}$ for $i \in \{1, 2, \cdots, T\}$.*

Theorem 1 tells us that it is much harder for SEAT to approximate the averaged prediction of history networks than the EMA method. So the prediction of SEAT keeps away from the averaged prediction of models, which suffers from homogenization. To ensure whether the difference in Theorem 1 is benign, we provide empirical evidence of the improvement of the self-ensemble technique on the loss landscape.

It is difficult for the traditional 1D linear interpolation method to visualize non-convexities (Goodfellow & Vinyals, 2015) and biases caused by invariance symmetries (e.g. batch normalization) in the DNN. Li et al. (2018); Wu et al. (2020) address these challenges, providing a scheme called filter normalization to explore the sharpness/flatness of DNNs on the loss landscape. Let $\boldsymbol{v}_1$ and $\boldsymbol{v}_2$ are two random direction vectors sampled from a random Gaussian distribution. So we plot the value of

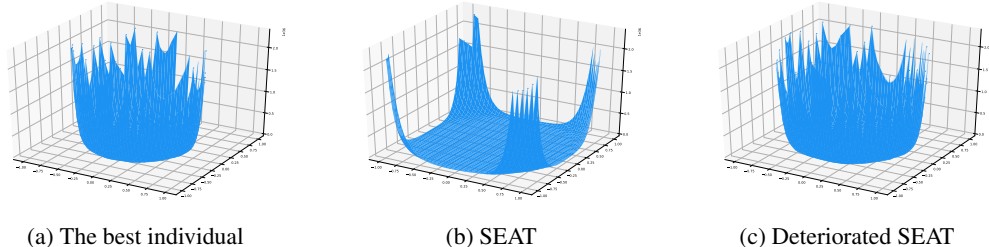

(a) The best individual        (b) SEAT        (c) Deteriorated SEAT

Figure 2: The loss surfaces of ResNet18 on CIFAR-10 test data. The loss surface of the standard adversarially trained models transitions up and down, which means the loss of such a model changes dramatically as moving along some directions. In contrast, SEAT has a fairly smooth landscape.

loss function around $\theta$ when inputting data samples:

$$L(\boldsymbol{\theta}; \boldsymbol{v}_1, \boldsymbol{v}_2) = \ell\left(\boldsymbol{\theta} + m\boldsymbol{v}_1 + n\boldsymbol{v}_2; x, y\right), \tag{9}$$

where $m = \frac{\|\boldsymbol{\theta}\|}{\|\boldsymbol{v}_1\|}$, $n = \frac{\|\boldsymbol{\theta}\|}{\|\boldsymbol{v}_2\|}$, and $\|\cdot\|$ denotes the Frobenius norm. Specifically, we apply the method to plot the surface of DNN within the 3D projected space. We visualize the loss values of ResNet18 on the testing dataset of CIFAR-10 in Figure 2. Each grid point in the x-axis represents a sampled gradient direction tuple $(\boldsymbol{v}_1, \boldsymbol{v}_2)$. It is clear that SEAT has a smooth and slowly varying loss function while the standard adversarially trained model visually has a sharper minimum with higher test error.

**Safeguard for the Initial Stage**: During the initial training period, the history models are less instrumental since they are not adequately learned. Therefore, it is not wise to directly make use of the ensembled weights, which may accumulate some naive weight states and do harm to approach the oracle. Therefore, we apply the protection mechanism for novice models at the beginning, denoted as $\alpha' = \min\left(\alpha, \frac{i}{i+c}\right)$, where $c$ is a constant factor controlling the length of warm-up periods for history models.

### 3.3 DETERIORATION: THE DEVIL IN THE LEARNING RATE

Having provided both theoretical and experimental support as well as a clear procedure of SEAT algorithm, we may take it for granted that this self-ensemble method could be integrated into the routine working flow as an off-shelf technique. But it is not the case.

As shown in Figure 2b and 2c, while the two self-ensemble models have *almost* nearly identical settings, their loss landscapes are completely different (quantitative results can be referred to Appendix A.4). Despite the fact that the landscape of deteriorated SEAT is a little mild than the standard adversarially trained model, it is still extremely steep when compared with SEAT. This phenomenon about deterioration is also discovered by Rebuffi et al. (2021). They combine model weight averaging with data augmentation and external data to handle the deterioration. The intention behind such a solution is robust overfitting (Rice et al., 2020) of individuals. However, it is unclear why an ensemble model assembled from overfitting individuals performs poorly, while inferior models can still be aggregated to produce a superior one in the middle of training stages. In the rest of this section, we provide another solution from a quite new perspective: *the strategy of learning rate*.

**Proposition 2.** *(Proof in Appendix B) Assuming that every candidate classifier is updated by SGD-like strategy, meaning $\boldsymbol{\theta}_{t+1} = \boldsymbol{\theta}_t - \tau_t \nabla_{\boldsymbol{\theta}_t} f_{\boldsymbol{\theta}_t}(x', y)$ with $\tau_1 \geq \tau_2 \geq \cdots \geq \tau_T > 0$, the performance of self-ensemble model depends on learning rate schedules.*

As illustrated in Figure 1a, we note that the phenomenon of homogenization exacerbates after each tipping point (e.g. Epoch 76, 91 and 111) when using the staircase strategy of learning rate. Combined with the trend of robust accuracy in Figure 1b, we find that the self-ensemble model cannot gain any benefits when the learning rate decays below a certain value. More results on different learning rate schedules can be found in Figure 3a.

## 4 EXPERIMENTAL RESULTS

In this section, we first conduct a set of experiments to verify the advantages of the proposed method. Then, we investigate how each component of SEAT functions and how it affects robust accuracy.

## 4.1 EXPERIMENTAL SETUP

We mainly use ResNet18 and WRN-32-10 (Zagoruyko & Komodakis, 2016) for the experiments on CIFAR-10/CIFAR-100 and all images are normalized into $[0, 1]$. For simplicity, we only report the results based on $L_\infty$ norm for the non-targeted attack. We train ResNet18 using SGD with 0.9 momentum for 120 epochs and the weight decay factor is set to $3.5e^{-3}$ for ResNet18 and $7e^{-4}$ for WRN-32-10. For SEAT, we use the piecewise linear learning rate schedule instead of the staircase one based on Proposition 2. The initial learning rate for ResNet18 is set to 0.01 and 0.1 for WRN-32-10 till Epoch 40 and then linearly reduced to 0.001, 0.0001 and 0.01, 0.001 at Epoch 60 and 120, respectively. The magnitude of maximum perturbation at each pixel is $\varepsilon = 8/255$ with step size $\kappa = 2/255$ and the PGD steps number in the inner maximization is 10. To evaluate adversarial defense methods, we apply several adversarial attacks including PGD (Madry et al., 2018), MIM (Dong et al., 2018), CW (Carlini & Wagner, 2017) and AutoAttack (AA) (Croce & Hein, 2020). We mainly compare the following defense methods in our experiments:

- **TRADES, MART, FAT and GAIRAT:** For TRADES and MART, we follow the official implementation of MART[§] to train both two models for better robustness. The hyper-parameter $\eta$ of both TRADES and MART is set to 6.0. The learning rate is divided by 10 at Epoch 75, 90, 100 for ResNet18 and at 75, 90, 110 for WRN-32-10, respectively. For FAT and GAIRAT, we completely comply with the official implementation[¶] to report results.

- **PoE:** We also perform the experiments on the Prediction-oriented Ensemble method (PoE). We select the above four advanced models as the candidate models. When averaging the output of these models, we assign different weights to individuals according to their robust accuracy. Specifically we respectively fix $\beta_{1,2,3,4} = 0.1, 0.2, 0.3, 0.4$ for FAT, GAIRAT, TRADES, and MART, named PoE. Considering the poor robust performance of FAT and GAIRAT, we also provide the PoE result by averaging the outputs of TRADES trained with $\lambda = 1, 2, 4, 6, 8$ and assign the same $\beta$ to them, named PoE (TRADES).

- **CutMix (with WA):** Based on the experimental results of Rebuffi et al. (2021), Cutmix with a fixed window size achieves the best robust accuracy. We follow this setting and set the window size to 20. Since we do not have matched computational resources with Deepmind, we only train models for 120 epochs with a batch size of 128 rather than 400 epochs with a batch size of 512. Besides, we neither use Swish/SiLU activation functions (Hendrycks & Gimpel, 2016) nor introduce external data and samples crafted by generative models (Brock et al., 2019; Ho et al., 2020; Child, 2021) for fair comparison.

## 4.2 ROBUSTNESS EVALUATION OF SEAT

In this section, we fully verify the effectiveness of the SEAT method. We report both average accuracy rates and standard deviations. All results in Tables 1 and 2 are computed with 5 individual trials. Results on ResNet18 are summarized in Table 1. Here, we mainly report the results on the CIFAR-10 dataset due to the space limitation. For the results on CIFAR-100, please refer to Appendix A.2. From the table, we can see that the superiority of SEAT is apparent especially considering we only use candidate models supervised by the traditional cross-entropy loss without external data. Compared with two advanced loss-oriented methods (TRADES and MART), the results of SEAT are at least $\sim 2\%$ better than TRADES and MART against CW and AA. We further emphasize that although the boost of SEAT regarding PGD attacks is slight when compared with GAIRAT, SEAT achieves startling robust accuracy results when facing CW and AA attacks. Note that AA is an ensemble of various advanced attacks (the performance under each component of AA can be found in Appendix A.1) and thus the robust accuracy of AA reliably reflects the adversarial robustness, which indicates that the robustness improvement of SEAT is not biased. It's worth noting that PoE is unexpectedly much weaker in defending AA attack even than its members. We guess the poor performance of FAT and GAIRAT encumber the collective though we appropriately lower the contribution score of them. If we switch to an ensemble of the outputs of TRADES with different $\lambda$s, the performance will be slightly improved. That demonstrates that the robustness of every candidate model will affect the performance of the ensemble. The negligible computational burden estimated in Appendix A.5 also demonstrates the superiority of our on-the-fly parameter-averaged method of SEAT.

We also evaluate the SEAT method on WRN-32-10. Results are shown in Table 2. We notice that the superiority of SEAT appears to be enhanced when using WRN-32-10. The gap between SEAT

---

[§]https://github.com/YisenWang/MART
[¶]https://github.com/zjfheart/Geometry-aware-Instance-reweighted-Adversarial-Training

Table 1: Comparison of our algorithm with different defense methods using ResNet18 on CIFAR-10. The maximum perturbation is $\varepsilon = 8/255$. Average accuracy rates (in %) and standard deviations have shown that the proposed SEAT method greatly improves the robustness of the model.

| Method | NAT | PGD$^{20}$ | PGD$^{100}$ | MIM | CW | AA |
|---|---|---|---|---|---|---|
| AT | 84.32±0.23 | 48.29±0.11 | 48.12±0.13 | 47.95±0.04 | 49.57±0.15 | 44.37±0.37 |
| TRADES | 83.91±0.33 | 54.25±0.11 | 52.21±0.09 | 55.65±0.1 | 52.22±0.05 | 48.2±0.2 |
| FAT | **87.72±0.14** | 46.69±0.31 | 46.81±0.3 | 47.03±0.17 | 49.66±0.38 | 43.14±0.43 |
| MART | 83.12±0.23 | 55.43±0.16 | 53.46±0.24 | **57.06±0.2** | 51.45±0.29 | 48.13±0.31 |
| GAIRAT | 83.4±0.21 | 54.76±0.42 | 54.81±0.63 | 53.57±0.31 | 38.71±0.26 | 31.25±0.44 |
| PoE | 85.41±0.29 | 55.2±0.37 | 55.07±0.24 | 54.33±0.32 | 49.25±0.16 | 46.17±0.35 |
| PoE (TRADES) | 83.57±0.31 | 53.88±0.45 | 53.82±0.27 | 55.01±0.18 | 52.72±0.65 | 49.2±0.24 |
| CutMix (with WA) | 81.26±0.44 | 52.77±0.33 | 52.55±0.25 | 53.01±0.25 | 50.01±0.55 | 47.38±0.36 |
| **SEAT** | 83.7±0.13 | **56.02±0.11** | **55.97±0.07** | **57.13±0.12** | **54.38±0.1** | **51.3±0.26** |
| **SEAT+CutMix** | 81.53±0.31 | 55.3±0.27 | 54.82±0.18 | 56.41±0.17 | 53.83±0.31 | 49.1±0.44 |

Table 2: Comparison of our algorithm with different defense methods using WRN-32-10 on CIFAR-10. The maximum perturbation is $\varepsilon = 8/255$. Average accuracy rates (in %) and standard deviations have shown that SEAT also shows a great improvement on robustness.

| Method | NAT | PGD$^{20}$ | PGD$^{100}$ | MIM | CW | AA |
|---|---|---|---|---|---|---|
| AT | 87.32±0.21 | 49.01±0.33 | 48.83±0.27 | 48.25±0.17 | 52.8±0.25 | 48.17±0.48 |
| TRADES | 85.11±0.77 | 54.58±0.49 | 54.82±0.38 | 55.67±0.31 | 54.91±0.21 | 52.19±0.44 |
| FAT | **89.65±0.04** | 48.74±0.23 | 48.69±0.18 | 48.24±0.16 | 52.11±0.71 | 46.7±0.4 |
| MART | 84.26±0.28 | 54.11±0.58 | 54.13±0.3 | 55.2±0.22 | 53.41±0.17 | 50.2±0.36 |
| GAIRAT | 85.92±0.69 | 58.51±0.42 | 58.48±0.34 | 58.37±0.27 | 44.31±0.22 | 39.64±1.01 |
| PoE | 87.1±0.25 | 55.75±0.2 | 55.47±0.19 | 56.04±0.31 | 53.66±0.18 | 49.44±0.35 |
| PoE (TRADES) | 86.03±0.37 | 54.26±0.47 | 54.73±0.21 | 55.01±0.22 | 55.52±0.18 | 53.2±0.4 |
| CutMix (with WA) | 82.79±0.44 | 58.43±1.21 | 58.2±0.83 | 58.95±0.57 | 58.32±0.43 | 54.1±0.82 |
| **SEAT** | 86.44±0.12 | 59.84±0.2 | 59.8±0.16 | **60.87±0.1** | 58.95±0.34 | 55.67±0.22 |
| **SEAT+CutMix** | 84.81±0.18 | **60.2±0.16** | **60.31±0.12** | 60.53±0.21 | **59.46±0.24** | **56.03±0.36** |

and advanced defense methods enlarges to at least $\sim 4\%$. In the setting with data augmentation, when compared with the results of ResNet18, SEAT combined with CutMix gains much higher performance on robust accuracy against several attacks as the model becomes larger. The observation given by Rebuffi et al. (2021) shows that additional generated data would overload the model with low capacity. When it comes to ours, we can further infer from a marginal and even a negative effect shown in Table 1 that the model with low capacity cannot benefit from both external and internal generated data. This phenomenon springs from the fact that CutMix techniques tend to produce augmented views that are far away from the original image they augment, which means that they are a little hard for small neural networks to learn.

## 4.3 ABLATION STUDY

In this section, we perform several ablation experiments to investigate how different aspects of SEAT influence its effectiveness. If not specified otherwise, the experiments are conducted on CIFAR-10 using ResNet18.

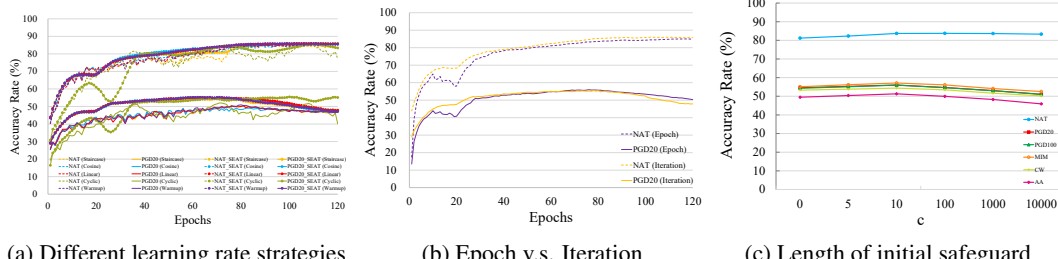

(a) Different learning rate strategies  (b) Epoch v.s. Iteration  (c) Length of initial safeguard

Figure 3: The overall ablation study of SEAT. The dashed line represents the natural accuracy while the solid line represents the robust accuracy.

**Different Learning Rate Strategies.** The staircase learning rate technique is always applied to update weights in commonly used neural networks, which $\tau$ remains unchanged within the specific

epochs and suddenly decreases to a certain value at some points. We also have the cosine, cyclic, and warming up learning rate schedules to compare with, in addition to the staircase and piecewise linear one deployed to SEAT. Figure 3a depicts the results of this experiment using various learning rate methodologies. The natural accuracy under various learning rate schedules is reported by the top dashed lines, which all grow gradually over time. After reaching their maxima, the relative bottom solid lines fall more or less. Quantitative Results of above five strategies from best checkpoints are shown in Appendix A.3. However, it can be observed that the red line (piecewise linear) and the blue line (cosine) decrease more slowly than the orange line (staircase) and they achieve higher robust accuracy at much later epochs than the staircase. Apparently, the piecewise linear strategy makes $\tau$ approach the threshold $\tau_{t'}$ at a slow pace while the cosine one increases the valid candidate models that can make contributions. Likewise, the cyclic strategy gets better robust accuracy and further improves the performance in the late stage. But the effect of warming up learning rate is marginal since the homogenization of the candidate models we mentioned in Sec. 3.2 cannot be fixed by the warmup strategy. These facts bear out Proposition 2 from the side.

**Epoch-based v.s. Iteration-based Ensembling.** Next, we investigate whether the frequency of ensembling has a great effect on the robustness of the collective. Specifically, we select two scenarios to compare with. One is an epoch-based ensemble that only gathers the updated models after traversing the whole training dataset. The other is an iteration-based ensemble collecting models whenever SGD functions. Comparison between the epoch-based and the iteration-based ensembling is shown in Figure 3b. It can be seen that the curves of both two schemes are nearly the same in their prime except that the iteration-based method has a slightly greater natural accuracy and degrades more slowly.

**Influence of the Initial Safeguard.** We also study the effect of the length of the initial safeguard. We plot all results against different kinds of attacks with $c$ ranging from 0 to 10000 (i.e. Epoch 32). The curves in Figure 3c show that the model ensembled with large $c$, which ignores more history models, results in weaker memorization and poor robustness. Actually, from the perspective of Proposition 2, $t'$ is fixed if the scheme of learning rate updating and the total training epochs are unchanged. So using a large $c$ is equivalent to squeeze the number of benign candidates before $t'$. Similarly, a small $c$ can manually filter the early models without being well trained. However, such a drop is minor because EMA has a built-in mechanism for adjusting.

**Mixed Strategies with SEAT for Improved Robustness.** Finally, we report the results under different mixed strategies on CIFAR-10 dataset with ResNet18 in Table 3. Overall, SEAT+TRADES and SEAT+TRADES+CutMix do not perform well. We believe the reason is that TRADES cannot well adapt to learning strategies other than the staircase one for its history snapshots of TRADES trained under other strategies are much inferior than the ones trained under the staircase learning rate strategy. However, MART is in harmony with other strategies. The combination of SEAT+MART+CutMix achieves the best or the second-best robustness against almost all types of attacks.

Table 3: Average robust accuracy (%) and standard deviation under different mixed strategies on CIFAR-10 dataset with ResNet18.

| Method | NAT | PGD20 | PGD100 | MIM | CW | APGDCE | APGDDLR | APGDT | FABT | Square | AA |
|---|---|---|---|---|---|---|---|---|---|---|---|
| SEAT | **83.7** | 56.02 | 55.97 | 57.13 | **54.38** | 53.87 | 53.35 | 50.88 | 51.41 | 57.77 | 51.3 |
| | **±0.13** | ±0.11 | ±0.07 | ±0.12 | **±0.1** | ±0.17 | ±0.24 | ±0.27 | ±0.37 | ±0.22 | ±0.26 |
| SEAT +TRADES | 81.21 | 57.05 | 57.0 | **57.92** | 52.75 | 50.75 | 50.36 | 48.56 | 49.45 | 54.45 | 49.91 |
| | ±0.44 | ±0.28 | ±0.15 | **±0.12** | ±0.09 | ±0.25 | ±0.29 | ±0.37 | ±0.65 | ±0.58 | ±0.17 |
| SEAT +MART | 78.94 | 56.92 | 56.88 | 57.24 | 52.09 | 54.06 | 53.77 | 51.03 | 51.11 | 57.76 | 51.7 |
| | ±0.14 | ±0.46 | ±0.29 | ±0.48 | ±0.66 | ±0.56 | ±0.25 | ±0.44 | ±0.41 | ±0.08 | ±0.35 |
| SEAT +TRADES+CutMix | 78.22 | 57.14 | 57.09 | 57.11 | 53.17 | 52.67 | 51.86 | 50.91 | 50.23 | 55.12 | 50.01 |
| | ±0.33 | ±0.21 | ±0.45 | ±0.39 | ±0.41 | ±0.82 | ±0.5 | ±0.17 | ±0.29 | ±0.46 | ±0.44 |
| SEAT +MART+CutMix | 75.87 | **57.3** | **57.29** | 57.47 | 53.13 | **54.33** | 53.98 | **51.2** | 51.42 | **58.01** | **52.1** |
| | ±0.24 | **±0.16** | **±0.43** | ±0.18 | ±0.2 | **±0.33** | **±0.61** | **±0.73** | **±0.17** | **±0.08** | **±0.22** |

## 5 CONCLUSION

In this paper, we propose a simple but powerful method called Self-Ensemble Adversarial Training (SEAT), which unites states of every history model on the optimization trajectory through the process of adversarial training. Compared with the standard ensemble method, SEAT only needs training once and has a better reusability. Besides, we give a theoretical explanation of the difference between the above two ensemble methods and visualize the change of loss landscape caused by SEAT. Furthermore, we analyze a subtle but fatal intrinsic issue in the learning rate strategy for the self-ensemble model, which causes the deterioration of the weight-ensembled method. Extensive experiments validate the effectiveness of the proposed SEAT method.

## ACKNOWLEDGEMENT

Yisen Wang is partially supported by the National Natural Science Foundation of China under Grant 62006153, Project 2020BD006 supported by PKU-Baidu Fund, and Open Research Projects of Zhejiang Lab (No. 2022RC0AB05).

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

# A  FURTHER EXPERIMENTS

Here we adopt ResNet18 and / or WRN-32-10 as the backbone model with the same experimental setup as in Sec. 4.1, where we reported the natural accuracy (NAT), PGD-20 and PGD-100 attack (PGD), MIM (PGD with a momentum term), CW attack and each component of AutoAttack. All the experiments are conducted for 5 individual trials and we also report their standard deviations. All the methods were realized by Pytorch 1.5, where we used a single NVIDIA GeForce RTX 3090 GPU.

## A.1  ROBUSTNESS AGAINST COMPONENTS OF AUTOATTACK

To broadly demonstrate the robustness of our proposal, we conducted experiments against each component of AutoAttack. We perform each component of AA on CIFAR-10 dataset with both ResNet18 and WRN-32-10, including three parameter-free versions of PGD with the CE, DLR, targeted-CE loss with 9 target classes loss ($APGD_{CE}$, $APGD_{DLR}$, $APGD_T$), the targeted version of FAB ($FAB_T$) and an existing complementary Square (Andriushchenko et al., 2020). Results are shown in the following Table 4. And it is obvious that our SEAT outperforms other methods against all components of AA.

Table 4: Average robust accuracy (%) and standard deviation against each component of AA on CIFAR-10 dataset with ResNet18 and WRN-32-10.

| | ResNet18 | | | | | WRN-32-10 | | | | |
| | $APGD_{CE}$ | $APGD_{DLR}$ | $APGD_T$ | $FAB_T$ | Square | $APGD_{CE}$ | $APGD_{DLR}$ | $APGD_T$ | $FAB_T$ | Square |
|---|---|---|---|---|---|---|---|---|---|---|
| AT | 47.47 ±0.35 | 48.57 ±0.18 | 45.14 ±0.31 | 46.17 ±0.11 | 54.21 ±0.15 | 49.17 ±0.26 | 50.09 ±0.36 | 47.34 ±0.33 | 48.00 ±0.43 | 56.5 ±0.18 |
| TRADES | 53.47 ±0.21 | 50.89 ±0.26 | 47.93 ±0.36 | 48.53 ±0.43 | 55.75 ±0.21 | 55.38 ±0.43 | 55.55 ±0.42 | 52.2 ±0.13 | 53.11 ±0.72 | 59.47 ±0.17 |
| MART | 52.98 ±0.13 | 50.36 ±0.3 | 48.17 ±0.72 | 49.39 ±0.28 | 55.73 ±0.51 | 55.2 ±0.32 | 55.41 ±0.4 | 51.99 ±0.3 | 52.88 ±0.63 | 59.01 ±0.38 |
| SEAT | **53.87** ±**0.17** | **53.35** ±**0.24** | **50.88** ±**0.27** | **51.41** ±**0.37** | **57.77** ±**0.22** | **57.57** ±**0.18** | **57.74** ±**0.29** | **55.06** ±**0.27** | **55.53** ±**0.36** | **62.26** ±**0.23** |

## A.2  PERFORMANCE ON CIFAR-100

To further demonstrate the robustness of our proposal against adversarial attacks, we benchmark the state-of-the-art robustness with ResNet18 on CIFAR-100. We widely investigate the performance of SEAT against the PGD methods ($PGD^{20}$ and $PGD^{100}$), MIM, CW, AA and its all components. Results shown in Table 5 demonstrate the effectiveness of SEAT for building a robust classifier.

Table 5: Comparison of our algorithm with different defense methods using ResNet18 on CIFAR10. The maximum perturbation is $\varepsilon = 8/255$. Average accuracy rates (in %) and standard deviations have shown that the proposed SEAT method greatly improves the robustness of the model.

| Method | NAT | $PGD^{20}$ | $PGD^{100}$ | MIM | CW | $APGD_{CE}$ | $APGD_{DLR}$ | $APGD_T$ | $FAB_T$ | Square | AA |
|---|---|---|---|---|---|---|---|---|---|---|---|
| AT | **60.1** ±**0.35** | 28.22 ±0.3 | 28.27 ±0.12 | 28.31 ±0.41 | 24.87 ±0.51 | 26.63 ±0.29 | 24.13 ±0.22 | 21.98 ±0.3 | 23.87 ±0.21 | 27.93 ±0.12 | 23.91 ±0.41 |
| TRADES | 59.93 ±0.46 | 29.9 ±0.41 | 29.88 ±0.11 | 29.55 ±0.25 | 26.14 ±0.21 | 27.93 ±0.44 | 25.43 ±0.29 | 23.72 ±0.45 | 25.16 ±0.15 | 30.03 ±0.32 | 24.72 ±0.37 |
| MART | 57.24 ±0.64 | 30.62 ±0.37 | 30.62 ±0.17 | 30.83 ±0.28 | 26.3 ±0.29 | 29.91 ±0.07 | 26.32 ±0.24 | 24.28 ±0.49 | 24.86 ±0.66 | 28.28 ±0.39 | 24.27 ±0.21 |
| **SEAT** | 56.28 ±0.33 | **32.15** ±**0.17** | **32.12** ±**0.26** | **32.62** ±**0.15** | **29.68** ±**0.26** | **30.97** ±**0.18** | **29.62** ±**0.22** | **26.88** ±**0.23** | **27.71** ±**0.24** | **32.35** ±**0.34** | **27.87** ±**0.24** |

## A.3  DIFFERENT LEARNING RATE STRATEGIES

Apart from showing the curve of different learning rate schedule in Figure 3 (a) in Sec. 4.3, we also report final results in Table 6. The effect of warming up learning rate is marginal. When compared with the staircase one, the warmup strategy cannot generate diverse models in the later stages so the homogenization of the candidate models we mentioned in Sec. 3.2 cannot be fixed by the warmup strategy. On the contrary, those methods like cosine / linear / cyclic that provide relatively diverse models in the later stages can mitigate the issue, accounting for more robust ensemble models.

## A.4  DETERIORATION OF VANILLA EMA

As shown in Sec. 3.3, the deteriorated SEAT underperforms SEAT a lot from the perspective of optimization. We also report quantitative results on both ResNet18 and WRN-32-10 shown in the

Table 6: Average robust accuracy (%) under different learning strategies on CIFAR-10 dataset with ResNet18.

| Method | NAT | $\text{PGD}^{20}$ | $\text{PGD}^{100}$ | MIM | CW | $\text{APGD}_{CE}$ | $\text{APGD}_{DLR}$ | $\text{APGD}_T$ | $\text{FAB}_T$ | Square | AA |
|---|---|---|---|---|---|---|---|---|---|---|---|
| SEAT (Staircase) | 80.91 | 54.58 | 54.56 | 54.47 | 49.71 | 52.39 | 48.01 | 45.83 | 45.11 | 53.64 | 45.85 |
| SEAT (Cosine) | 83.0 | 55.09 | 55.16 | 56.39 | 53.43 | 52.34 | 52.1 | 49.51 | 50.15 | 56.48 | 50.48 |
| SEAT (Linear) | **83.7** | **56.02** | **55.97** | **57.13** | **54.38** | **53.87** | **53.35** | **50.88** | **51.41** | **57.77** | **51.3** |
| SEAT (Warmup) | 82.74 | 55.31 | 55.35 | 56.39 | 53.26 | 53.55 | 48.94 | 45.89 | 46.6 | 54.94 | 45.82 |
| SEAT (Cyclic) | 83.14 | **56.03** | 55.79 | 56.99 | **54.01** | 53.72 | **53.1** | 50.66 | **51.02** | 57.75 | **51.44** |

Tables 7 and 8. The deteriorated one does not bring too much boost when compared to the vanilla adversarial training (except for PGD methods). A plausible explanation for the exception of PGD is that the SEAT technique produces an ensemble of individuals that are adversarially trained by PGD with the cross-entropy loss, which means that they are intrinsically good at defending the PGD attack with the cross-entropy loss and its variants even though they suffer from the deterioration. Considering results have greatly improved after using the piecewise linear learning rate strategy, it is fair to say that adjusting learning rate is effective. As we claimed in Proposition 2 and its proof, the staircase will inevitably make the self-ensemble model worsen since $\sum_{t=1}^{T}(\beta_t \tilde{\boldsymbol{\xi}}^\top)$ will gradually approach to zero, meaning the difference between $\bar{f}_{\mathcal{F}}(x, y)$ and $f_{\hat{\boldsymbol{\theta}}}(x, y)$ achieves the second order of smallness.

Table 7: Average robust accuracy (%) and standard deviation on CIFAR-10 dataset with ResNet18.

| Method | NAT | $\text{PGD}^{20}$ | $\text{PGD}^{100}$ | MIM | CW | $\text{APGD}_{CE}$ | $\text{APGD}_{DLR}$ | $\text{APGD}_T$ | $\text{FAB}_T$ | Square | AA |
|---|---|---|---|---|---|---|---|---|---|---|---|
| AT | **84.32** | 48.29 | 48.12 | 47.95 | 49.57 | 47.47 | 48.57 | 45.14 | 46.17 | 54.21 | 44.37 |
| | ±**0.23** | ±0.11 | ±0.13 | ±0.04 | ±0.15 | ±0.35 | ±0.18 | ±0.31 | ±0.11 | ±0.25 | ±0.37 |
| SEAT (deteriorated) | 80.91 | 54.58 | 54.56 | 54.47 | 49.71 | 52.39 | 48.01 | 45.83 | 45.11 | 53.64 | 45.85 |
| | ±0.38 | ±0.71 | ±0.29 | ±0.39 | ±0.41 | ±0.26 | ±0.18 | ±0.52 | ±0.23 | ±0.44 | ±0.19 |
| SEAT | 83.7 | **56.02** | **55.97** | **57.13** | **54.38** | **53.87** | **53.35** | **50.88** | **51.41** | **57.77** | **51.3** |
| | ±0.13 | ±**0.11** | ±**0.07** | ±**0.12** | ±**0.1** | ±**0.17** | ±**0.24** | ±**0.27** | ±**0.37** | ±**0.22** | ±**0.26** |

Table 8: Average robust accuracy (%) and standard deviation on CIFAR-10 dataset with WRN-32-10.

| Method | NAT | $\text{PGD}^{20}$ | $\text{PGD}^{100}$ | MIM | CW | $\text{APGD}_{CE}$ | $\text{APGD}_{DLR}$ | $\text{APGD}_T$ | $\text{FAB}_T$ | Square | AA |
|---|---|---|---|---|---|---|---|---|---|---|---|
| AT | **87.32** | 49.01 | 48.83 | 48.25 | 52.8 | 54.17 | 53.09 | 48.34 | 49.00 | 57.5 | 48.17 |
| | ±**0.21** | ±0.33 | ±0.27 | ±0.17 | ±0.25 | ±0.26 | ±0.36 | ±0.33 | ±0.43 | ±0.18 | ±0.48 |
| SEAT (deteriorated) | 85.28 | 55.68 | 55.57 | 55.6 | 53.01 | 54.12 | 53.54 | 49.95 | 50.02 | 57.81 | 49.96 |
| | ±0.42 | ±0.42 | ±0.19 | ±0.23 | ±0.41 | ±0.54 | ±0.28 | ±0.67 | ±0.75 | ±0.33 | ±0.31 |
| **SEAT** | 86.44 | **59.84** | **59.8** | **60.87** | **58.95** | **57.57** | **57.74** | **55.06** | **55.53** | **62.26** | **55.67** |
| | ±0.12 | ±**0.2** | ±**0.16** | ±**0.1** | ±**0.34** | ±**0.18** | ±**0.29** | ±**0.27** | ±**0.36** | ±**0.23** | ±**0.22** |

## A.5 COMPUTATIONAL COMPLEXITY FOR SEAT

To demonstrate the efficiency of the SEAT method, we use the number of Multiply-Accumulate operations (MACs) in Giga (G) to compute the theoretical amount of multiply-add operations in DNNs, roughly GMACs = 0.5 * GFLOPs. Besides, we also provide the actual running time. As shown in Table 9, the SEAT method takes negligible MACs and training time when compared with standard adversarial training.

Table 9: Evaluation of time complexity of SEAT. Here we use the number of Multiply-Accumulate operations (MACs) in Giga (G) to measure the running time complexity. And we also compute the actual training time with or without the SEAT method using ResNet18 and WRN-32-10 on a single NVIDIA GeForce RTX 3090 GPU.

| Method | MACs (G) | Training Time (mins) |
|---|---|---|
| ResNet18 (AT) | 0.56 | 272 |
| ResNet18 (SEAT) | 0.59 | 273 |
| WRN-32-10 (AT) | 6.67 | 1534 |
| WRN-32-10 (SEAT) | 6.81 | 1544 |

## B    PROOFS OF THEORETICAL RESULTS

### B.1    PROOF OF PROPOSITION 1

**Proposition 1.** *(Restated) Let $f_{\boldsymbol{\theta}}(\cdot)$ denote the predictions of a neural network parametrized by weights $\boldsymbol{\theta}$. Assuming that $\forall \boldsymbol{\theta} \in \Theta$, $f_{\boldsymbol{\theta}}(\cdot)$ is continuous and $\forall (x, y) \in \mathbb{D}$, $f_{\boldsymbol{\theta}}(x, y)$ is at least twice differentiable. Consider two points $\boldsymbol{\theta}_t, \tilde{\boldsymbol{\theta}} \in \Theta$ in the weight space and let $\boldsymbol{\xi} = \boldsymbol{\theta}_t - \tilde{\boldsymbol{\theta}}$, for $t \in \{1, 2, \cdots, T\}$, the difference between $\bar{f}_{\mathcal{F}}(x, y)$ and $f_{\tilde{\boldsymbol{\theta}}}(x, y)$ is of the second order of smallness if and only if $\sum_{t=1}^{T}(\beta_t \boldsymbol{\xi}^\top) = \mathbf{0}$.*

*Proof.* For the sake of the twice differentiability of $f_{\boldsymbol{\theta}}(x, y)$, based on the Taylor expansion, we can fit a quadratic polynomial of $f_{\tilde{\boldsymbol{\theta}}}(x, y)$ to approximate the value of $f_{\boldsymbol{\theta}_t}(x, y)$:

$$f_{\boldsymbol{\theta}_t}(x, y) = f_{\tilde{\boldsymbol{\theta}}}(x, y) + \boldsymbol{\xi}^\top \nabla_{\boldsymbol{\xi}} f_{\tilde{\boldsymbol{\theta}}}(x, y) + \frac{1}{2} \boldsymbol{\xi}^\top \nabla_{\boldsymbol{\xi}}^2 f_{\tilde{\boldsymbol{\theta}}}(x, y) \boldsymbol{\xi} + O\left(\Delta^n\right), \tag{10}$$

where $O\left(\Delta^n\right)$ represents the higher-order remainder term. Note that the subscript $\boldsymbol{\xi}$ here stands for a neighborhood where the Taylor expansion approximates a function by polynomials of any point (i.e. $\tilde{\boldsymbol{\theta}}$) in terms of its value and derivatives. So the difference between the averaged prediction of candidate classifiers and the prediction of the ensembled weight classifier can be formulated as:

$$\begin{aligned}
\bar{f}_{\mathcal{F}}(x, y) - f_{\tilde{\boldsymbol{\theta}}}(x, y) &= \sum_{t=1}^{T} \beta_t f_{\boldsymbol{\theta}_t}(x, y) - f_{\tilde{\boldsymbol{\theta}}}(x, y) \\
&= \sum_{t=1}^{T} \beta_t f_{\tilde{\boldsymbol{\theta}}}(x, y) + \sum_{t=1}^{T} \beta_t \boldsymbol{\xi}^\top \nabla_{\boldsymbol{\xi}} f_{\tilde{\boldsymbol{\theta}}}(x, y) + \sum_{t=1}^{T} \beta_t O\left(\Delta^2\right) - f_{\tilde{\boldsymbol{\theta}}}(x, y) \quad (11) \\
&= \sum_{t=1}^{T} (\beta_t \boldsymbol{\xi}^\top) \nabla_{\boldsymbol{\xi}} f_{\tilde{\boldsymbol{\theta}}}(x, y) + O(\Delta^2).
\end{aligned}$$

Therefore, we can claim that the difference between $f_{\boldsymbol{\theta}_t}(x, y)$ and $f_{\tilde{\boldsymbol{\theta}}}(x, y)$ is "almost" at least of the first order of smallness except for some special cases. And we will immediately declare under which condition this difference can achieve the second order of smallness in the following proof of Theorem 1. $\square$

### B.2    PROOF OF THEOREM 1

**Theorem 1.** *(Restated) Assuming that for $i, j \in \{1, \cdots, T\}$, $\boldsymbol{\theta}_i = \boldsymbol{\theta}_j$ if and only if $i = j$. The difference between the averaged prediction of multiple networks and the prediction of SEAT is of the second order of smallness if and only if $\beta_i = (1 - \alpha)^{1 - \delta(i-1)} \alpha^{T-i}$ for $i \in \{1, 2, \cdots, T\}$.*

*Proof.* According to Eqn 11, we know that the second order of smallness will achieve when $\sum_{i=1}^{T}(\beta_i \boldsymbol{\xi}^\top) = \mathbf{0}$. Thus, we continue deducing from Eqn 11 as:

$$\begin{aligned}
\sum_{i=1}^{T}(\beta_i \boldsymbol{\xi}^\top) &= \mathbf{0} \\
\sum_{i=1}^{T} \beta_i (\boldsymbol{\theta}_i - \tilde{\boldsymbol{\theta}}) &= \mathbf{0} \\
\sum_{i=1}^{T} \beta_i \boldsymbol{\theta}_i &= \tilde{\boldsymbol{\theta}} \\
\sum_{i=1}^{T} \beta_i \boldsymbol{\theta}_i &= \sum_{i=1}^{T} (1 - \alpha)^{1 - \delta(i-1)} \alpha^{T-i} \boldsymbol{\theta}_i.
\end{aligned} \tag{12}$$

To get a further conclusion, we next use Mathematical Induction (MI) to prove only when $\beta_i = (1 - \alpha)^{1 - \delta(i-1)} \alpha^{T-i}$ for $i \in \{1, 2, \cdots, T\}$ will lead to $\sum_{i=1}^{T} \beta_i \boldsymbol{\theta}_i = \sum_{i=1}^{T} (1 - \alpha)^{1 - \delta(i-1)} \alpha^{T-i} \boldsymbol{\theta}_i$.

**Base case:** Let $i = 1$, it is clearly true that $\beta_1 = \alpha^{T-1}$ if and only if $\beta_1 \boldsymbol{\theta}_1 = \alpha^{T-1} \boldsymbol{\theta}_1$, hence the base case holds.

**Inductive step:** Assume the induction hypothesis that for a particular $k$, the single case $T = k$ holds, meaning the sequence of $(\beta_1, \beta_2, \cdots, \beta_k)$ is equal to the sequence of $((1 - \alpha)^{1-\delta(0)}\alpha^{T-1}, (1 - \alpha)^{1-\delta(1)}\alpha^{T-2}, \cdots, (1 - \alpha)^{1-\delta(k-1)}\alpha^{T-k})$ if $\sum_{i=1}^{k} \beta_i \boldsymbol{\theta}_i = \sum_{i=1}^{k}(1 - \alpha)^{1-\delta(i-1)}\alpha^{T-i}\boldsymbol{\theta}_i$.

For $T = k + 1$, it follows that:

$$\sum_{i=1}^{k+1} \beta_i \boldsymbol{\theta}_i = \sum_{i=1}^{k+1}(1 - \alpha)^{1-\delta(i-1)}\alpha^{T-i}\boldsymbol{\theta}_i$$

$$\sum_{i=1}^{k} \beta_i \boldsymbol{\theta}_i + \beta_{k+1}\boldsymbol{\theta}_{k+1} = \sum_{i=1}^{k}(1 - \alpha)^{1-\delta(i-1)}\alpha^{T-i}\boldsymbol{\theta}_i + (1 - \alpha)^{1-\delta((k+1)-1)}\alpha^{T-(k+1)}\boldsymbol{\theta}_{k+1} \quad (13)$$

$$\beta_{k+1}\boldsymbol{\theta}_{k+1} = (1 - \alpha)^{1-\delta((k+1)-1)}\alpha^{T-(k+1)}\boldsymbol{\theta}_{k+1}$$

$$\beta_{k+1} = (1 - \alpha)^{1-\delta((k+1)-1)}\alpha^{T-(k+1)}.$$

The sequence of normalized scores at the $(k+1)$-th ensembling at left hand is $(\beta_1, \beta_2, \cdots, \beta_k, \beta_{k+1})$ after adding the new term $\beta_{k+1}$. Likewise, the sequence of the right hand is $(\beta_1, \beta_2, \cdots, \beta_k)$ is equal to the sequence of $((1 - \alpha)^{1-\delta(0)}\alpha^{T-1}, (1 - \alpha)^{1-\delta(1)}\alpha^{T-2}, \cdots, (1 - \alpha)^{1-\delta(k-1)}\alpha^{T-k})$. Because every $f_{\theta_t} \in \mathcal{F}$ is different from others and the sequence is ordered, we have $(\beta_1, \beta_2, \cdots, \beta_k, \beta_{k+1}) = ((1 - \alpha)^{1-\delta(0)}\alpha^{T-1}, (1 - \alpha)^{1-\delta(1)}\alpha^{T-2}, \cdots, (1 - \alpha)^{1-\delta((k+1)-1)}\alpha^{T-(k+1)})$.

**Conclusion:** Since both the base case and the inductive step have been proved as true, by mathematical induction the statement $\beta_i = (1 - \alpha)^{1-\delta(i-1)}\alpha^{T-i}$ for $i \in \{1, 2, \cdots, T\}$ holds for every positive integer $T$. Following Eqn 11, the difference between SEAT and the averaged prediction of history networks is controlled by the first order term which reaches 0 only at $\beta_i = (1 - \alpha)^{1-\delta(i-1)}\alpha^{T-i}$ for $i \in \{1, 2, \cdots, T\}$. Thus, SEAT is hardly be approximate to the averaged prediction of history networks indeed. □

### B.3 PROOF OF PROPOSITION 2

**Proposition 2.** *(Restated) Assuming that every candidate classifier is updated by SGD-like strategy, meaning $\boldsymbol{\theta}_{t+1} = \boldsymbol{\theta}_t - \tau_t \nabla_{\boldsymbol{\theta}_t} f_{\boldsymbol{\theta}_t}(x', y)$ with $\tau_1 \geq \tau_2 \geq \cdots \geq \tau_T > 0$, the performance of self-ensemble model depends on learning rate schedules.*

*Proof.* First we discuss a special case - the change at the $t$-th iteration. Reconsidering the first order term in Eqn 11, we have:

$$
\sum_{t=1}^{T}(\beta_t \boldsymbol{\xi}^\top)\nabla_{\boldsymbol{\xi}} f_{\tilde{\boldsymbol{\theta}}}(x, y)
$$

$$
= \sum_{t=1}^{T}[\beta_t(\boldsymbol{\theta}_t - \tilde{\boldsymbol{\theta}})]\nabla_{\boldsymbol{\xi}} f_{\tilde{\boldsymbol{\theta}}}(x, y)
$$

$$
= \sum_{t=1}^{T}[\beta_t((1 - (1-\alpha)^{1-\delta(t-1)}\alpha^{T-t})\boldsymbol{\theta}_t - \tilde{\boldsymbol{\theta}}_{\mathcal{F}\backslash t})]\nabla_{\boldsymbol{\xi}} f_{\tilde{\boldsymbol{\theta}}}(x, y)
$$

$$
= \sum_{t=1}^{T}[\beta_t((1 - (1-\alpha)^{1-\delta(t-1)}\alpha^{T-t})\boldsymbol{\theta}_t - (1 - (1-\alpha)^{1-\delta(t-1)}\alpha^{T-t})\boldsymbol{\theta}_{t-1} + \boldsymbol{\theta}_{t-1}
$$
$$
- (1-\alpha)^{1-\delta(t-1)}\alpha^{T-t}\boldsymbol{\theta}_{t-1} - \tilde{\boldsymbol{\theta}}_{\mathcal{F}\backslash t})]\nabla_{\boldsymbol{\xi}} f_{\tilde{\boldsymbol{\theta}}}(x, y)
$$

$$
= \sum_{t=1}^{T}[\beta_t((1 - (1-\alpha)^{1-\delta(t-1)}\alpha^{T-t})\boldsymbol{\theta}_t - (1 - (1-\alpha)^{1-\delta(t-1)}\alpha^{T-t})\boldsymbol{\theta}_{t-1} + \boldsymbol{\theta}_{t-1}
$$
$$
- (1-\alpha)^{1-\delta(t-1)}\alpha^{T-t}\boldsymbol{\theta}_{t-1} - (1-\alpha)^{1-\delta(t-1)}\alpha^{T-t+1}\boldsymbol{\theta}_{t-1} - \tilde{\boldsymbol{\theta}}_{\mathcal{F}\backslash t, t-1})]\nabla_{\boldsymbol{\xi}} f_{\tilde{\boldsymbol{\theta}}}(x, y)
$$

$$
= \sum_{t=1}^{T}[\beta_t((1 - (1-\alpha)^{1-\delta(t-1)}\alpha^{T-t})\boldsymbol{\theta}_t - (1 - (1-\alpha)^{1-\delta(t-1)}\alpha^{T-t})\boldsymbol{\theta}_{t-1}
$$
$$
+ (1 - (1-\alpha)^{1-\delta(t-1)}\alpha^{T-t} - (1-\alpha)^{1-\delta(t-1)}\alpha^{T-t}\alpha)\boldsymbol{\theta}_{t-1} - \tilde{\boldsymbol{\theta}}_{\mathcal{F}\backslash t, t-1})]\nabla_{\boldsymbol{\xi}} f_{\tilde{\boldsymbol{\theta}}}(x, y),
\tag{14}
$$

so we can deduce by combining:

$$
\sum_{t=1}^{T}[\beta_t((1 - (1-\alpha)^{1-\delta(t-1)}\alpha^{T-t})(\boldsymbol{\theta}_t - \boldsymbol{\theta}_{t-1}) + C)]\nabla_{\boldsymbol{\xi}} f_{\tilde{\boldsymbol{\theta}}}(x, y),
\tag{15}
$$

where $C = (1 - (1-\alpha)^{1-\delta(t-1)}\alpha^{T-t} - (1-\alpha)^{1-\delta(t-1)}\alpha^{T-t}\alpha)\boldsymbol{\theta}_{t-1} - \tilde{\boldsymbol{\theta}}_{\mathcal{F}\backslash t, t-1}$. By using SGD to update $\theta_t$, we have:

$$
\sum_{t=1}^{T}[\beta_t((1 - (1-\alpha)^{1-\delta(t-1)}\alpha^{T-t})(\tau_t \mathbb{E}_{(x,y)}(\nabla_{\boldsymbol{\theta}_t}\ell(\boldsymbol{\theta}_t; (x'_k, y)) + C))]\nabla_{\boldsymbol{\xi}} f_{\tilde{\boldsymbol{\theta}}}(x, y).
\tag{16}
$$

Considering $C$ is a constant for the $t$-th update, without changing samples in the $t$-th minibatch, we can conclude that the output of SEAT depends on the learning rate $\tau_t$.

To further analyse the whole training process, we construct $\boldsymbol{\theta} = \frac{1}{T}\sum_{t=1}^{T}\boldsymbol{\theta}_t$ and $\tilde{\boldsymbol{\xi}} = \boldsymbol{\theta} - \tilde{\boldsymbol{\theta}}$ to unpack $\tilde{\boldsymbol{\theta}}_{\mathcal{F}\backslash t}$ for the averaged prediction of history networks, and then reformulate Eqn 11:

$$\sum_{t=1}^{T}(\beta_t\tilde{\boldsymbol{\xi}}^{\top})\nabla_{\tilde{\boldsymbol{\xi}}}f_{\tilde{\boldsymbol{\theta}}}(x,y)$$

$$= \sum_{t=1}^{T}[\beta_t(\boldsymbol{\theta} - \tilde{\boldsymbol{\theta}})]\nabla_{\tilde{\boldsymbol{\xi}}}f_{\tilde{\boldsymbol{\theta}}}(x,y)$$

$$= \sum_{t=1}^{T}[\beta_t((\frac{1}{T} - (1-\alpha)^{1-\delta(t-1)}\alpha^{T-t})\boldsymbol{\theta}_t - \tilde{\boldsymbol{\theta}}_{\mathcal{F}\backslash t})]\nabla_{\tilde{\boldsymbol{\xi}}}f_{\tilde{\boldsymbol{\theta}}}(x,y)$$

$$= \sum_{t=1}^{T}[\beta_t((\frac{1}{T} - (1-\alpha)^{1-\delta(t-1)}\alpha^{T-t})\boldsymbol{\theta}_t - (\frac{1}{T} - (1-\alpha)^{1-\delta(t-1)}\alpha^{T-t})\boldsymbol{\theta}_{t-1} + \frac{\boldsymbol{\theta}_{t-1}}{T}$$
$$- (1-\alpha)^{1-\delta(t-1)}\alpha^{T-t}\boldsymbol{\theta}_{t-1} - \tilde{\boldsymbol{\theta}}_{\mathcal{F}\backslash t})]\nabla_{\tilde{\boldsymbol{\xi}}}f_{\tilde{\boldsymbol{\theta}}}(x,y)$$

$$= \sum_{t=1}^{T}[\beta_t((\frac{1}{T} - (1-\alpha)^{1-\delta(t-1)}\alpha^{T-t})\boldsymbol{\theta}_t - (\frac{1}{T} - (1-\alpha)^{1-\delta(t-1)}\alpha^{T-t})\boldsymbol{\theta}_{t-1} + \frac{\boldsymbol{\theta}_{t-1}}{T}$$
$$- (1-\alpha)^{1-\delta(t-1)}\alpha^{T-t}\boldsymbol{\theta}_{t-1} - (1-\alpha)^{1-\delta(t-1)}\alpha^{T-t+1}\boldsymbol{\theta}_{t-1} - \tilde{\boldsymbol{\theta}}_{\mathcal{F}\backslash t,t-1})]\nabla_{\tilde{\boldsymbol{\xi}}}f_{\tilde{\boldsymbol{\theta}}}(x,y)$$

$$= \sum_{t=1}^{T}[\beta_t((\frac{1}{T} - (1-\alpha)^{1-\delta(t-1)}\alpha^{T-t})\boldsymbol{\theta}_t - (\frac{1}{T} - (1-\alpha)^{1-\delta(t-1)}\alpha^{T-t})\boldsymbol{\theta}_{t-1} + (\frac{1}{T}$$
$$- (1-\alpha)^{1-\delta(t-1)}\alpha^{T-t} - (1-\alpha)^{1-\delta(t-1)}\alpha^{T-t}\alpha)\boldsymbol{\theta}_{t-1} - \tilde{\boldsymbol{\theta}}_{\mathcal{F}\backslash t,t-1})]\nabla_{\tilde{\boldsymbol{\xi}}}f_{\tilde{\boldsymbol{\theta}}}(x,y)$$

$$= \sum_{t=1}^{T}[\beta_t((\frac{1}{T} - (1-\alpha)^{1-\delta(t-1)}\alpha^{T-t})(\boldsymbol{\theta}_t - \boldsymbol{\theta}_{t-1}) + C')]\nabla_{\boldsymbol{\xi}}f_{\tilde{\boldsymbol{\theta}}}(x,y),$$

(17)

where $C' = (\frac{1}{T} - (1-\alpha)^{1-\delta(t-1)}\alpha^{T-t} - (1-\alpha)^{1-\delta(t-1)}\alpha^{T-t}\alpha)\boldsymbol{\theta}_{t-1} - \tilde{\boldsymbol{\theta}}_{\mathcal{F}\backslash t,t-1}$. Likewise, the above equation can be further deduced:

$$\sum_{t=1}^{T}[\beta_t((\frac{1}{T} - (1-\alpha)^{1-\delta(t-1)}\alpha^{T-t})(\tau_t\mathbb{E}_{(x,y)}(\nabla_{\boldsymbol{\theta}_t}\ell(\boldsymbol{\theta}_t;(x'_k,y)) + C'))]\nabla_{\boldsymbol{\xi}}f_{\tilde{\boldsymbol{\theta}}}(x,y), \quad (18)$$

which means the difference between $\bar{f}_{\mathcal{F}}(x,y)$ and $f_{\tilde{\boldsymbol{\theta}}}(x,y)$ depends on learning rate schedules. $\quad\square$

