# OpenReview forum: "Self-ensemble Adversarial Training for Improved Robustness"
_ICLR.cc/2022/Conference — ICLR 2022 Poster_

### Official Review · Reviewer_E4Bu · 2021-11-02

**Correctness:** 3
**Technical Novelty And Significance:** 2
**Empirical Novelty And Significance:** 2
**Recommendation:** 6
**Confidence:** 4

**Main Review:**

Strengths:
* The proposed method is well-motived, and authors provided multiple experiments to support their claim and justify the performances of their proposed method.
* This paper is overall clear, well-organized and easy to follow.

Weaknesses / discussion questions:
* Deterioration is a very interesting topic and desired more studies. The paper didn't explain in detail what caused deteriorated SEAT (in Figure 2). And does deteriorated SEAT underperform SEAT a lot? If the answer is yes, does adjusting learning rate sufficiently enough to prevent this decreasing? I believe the paper could be improved if a clear explanation can be studied and explored.

* The experimental results are not convincing to me. My specific concerns are the following:
  * The authors are suggested to report more details for experimental settings, e.g., whether report **best** or the **last** epoch results? what loss function used for SEAT? if xentropy used, then can the performance be further boosted if using TRADES? The authors are suggested to report more ablation study discussing different combinations (e.g., SEAT by TRADES, SEAT + CutMix by TRADES) as well.
  * It does not make sense to me that PoE involves with FAT and GAIRAT, especially when TABLE 1 already provides evidence that FAT and GAIRAT fails on robustness. It would be more interesting and meaningful when including stronger components, e.g., TRADES trained with different \lambda.
  * On Table 1, I found at least TRADES is not sufficiently trained, its performance over NAT and AA should be at least better than 84% and 49.5% respectively. The authors are suggested to re-train and re-evaluate the baselines.
  * On Table 2, authors claim that "when compared with the results of ResNet18, CutMix performs much better when the model becomes larger." However, SEAT+CutMix didn't achieve the best against all the attacks (e.g., MIM), and we can also observe SEAT (86.44) performs better than SEAT+CutMix (84.81) over NAT, so CutMix's better performance over AA could be due to the trade-off between Natural accuracy and robustness. It is well possible that CutMix may benefit the model with more capacity, but this conclusion cannot be directly drawn based on the current observations on Table 2.
  * For a thorough evaluation, it would be better to report larger datasets, e.g., CIFAR-100.
  * Given the results reported for the proposed method show a great boost in robust performance, it would be nice to have the ability to view the training and evaluation code.

* Although the paper is overall easy to follow, it seems to be written in a rush. There are several typos. The writing of the paper needs further polishing.
  * Page 5, Figure 1(b): Resnet -> ResNet
  * Page 7, first row under section 4.1: wrong citation format
  * Page 8, Paragraph 1: robusta -> robust



**Summary Of The Paper:**

This paper proposes an adversarial training scheme called SEAT, where the final robust classifier is united by averaging states of every history models through the adversarial training process. Authors compares SEAT with prediction-ensembled methods and analyze its effectiveness both theoretically and experimentally.

**Summary Of The Review:**

This paper proposes an adversarial training scheme called SEAT, where the final robust classifier is united by averaging states of every history models through the adversarial training process. Although the idea of self-weight-ensembled method is intriguing, I am not that convinced by the current experimental settings and found it is lack of several competing baselines and performance on larger dataset.

---

> ### Author Response · Authors · 2021-11-19
> **Response to reviewer E4Bu (1/4)**
>
> Thanks for your reviews and comments on the paper. We found the set of perceptive questions you raised in your feedback very constructive, pushing us to think of a more comprehensive experiment design. We provide pointwise responses below and all the typos have been fixed in the revision.
>
> Q1: Does deteriorated SEAT underperform SEAT a lot? Does adjusting the learning rate sufficiently enough to prevent this decreasing?
>
> A1: Yes, the deteriorated SEAT underperforms SEAT a lot and we post the results of ResNet18 and WRN-32-10 in Tables S14 and S15. The deteriorated one does not bring too much boost when compared to the vanilla adversarial training (except for PGD methods). A plausible explanation for the exception of PGD is that the SEAT technique produces an ensemble of individuals that are adversarially trained by PGD with the cross-entropy loss, which means that they are intrinsically good at defending the PGD attack with the cross-entropy loss and its variants even though they suffer from the deterioration. Considering results have greatly improved after using the piecewise linear learning rate strategy, it is fair to say that adjusting the learning rate is effective. As we claimed in Proposition 2 and its proof, the staircase will inevitably make the self-ensemble model worsen since $\sum_{t=1}^{T} (\beta_{t}\tilde{\boldsymbol\xi}^{\mathrm{T}})$​ will gradually approach to zero, meaning the difference between $\bar{f}_{\mathcal{F}}(x,y)$​ and $f_{\tilde{\boldsymbol\theta}}(x, y)$​ achieves the second order of smallness. Nevertheless, it should not be the only way to solve the homogenization issue of the self-ensemble method. Importing external data through the training process may be another way to handle the problem [1], yet it needs a mass amount of computational resources.
>
> Table S14: Average robust accuracy (%) and standard deviation on CIFAR-10 dataset with ResNet18.
>
> | Method      |          NAT          |   $\operatorname{PGD}^{20}$     |   $\operatorname{PGD}^{100}$    |          MIM          |          CW          |   $\operatorname{APGD}_{CE}$    |  $\operatorname{APGD}_{DLR}$   |   $\operatorname{APGD}_{T}$    |    $\operatorname{FAB}_{T}$    |        Square         |          AA          |
> | :---------- | :-------------------: | :-------------------: | :-------------------: | :-------------------: | :------------------: | :-------------------: | :-------------------: | :-------------------: | :-------------------: | :-------------------: | :------------------: |
> | AT                  | **84.32&plusmn;0.23** |   48.29&plusmn;0.11   |   48.12&plusmn;0.13   |   47.95&plusmn;0.04   |  49.57&plusmn;0.15   |   47.47&plusmn;0.35   |   48.57&plusmn;0.18   |   45.14&plusmn;0.31   |   46.17&plusmn;0.11   |   54.21&plusmn;0.25   |  44.37&plusmn;0.37   |
> | SEAT (deteriorated) |   80.91&plusmn;0.38   |   54.58&plusmn;0.71   |   54.56&plusmn;0.29   |   54.47&plusmn;0.39   |  49.71&plusmn;0.41   |   52.39&plusmn;0.26   |   48.01&plusmn;0.18   |   45.83&plusmn;0.52   |   45.11&plusmn;0.23   |   53.64&plusmn;0.44   |  45.85&plusmn;0.19   |
> | **SEAT**            |   83.7&plusmn;0.13    | **56.02&plusmn;0.11** | **55.97&plusmn;0.07** | **57.13&plusmn;0.12** | **54.38&plusmn;0.1** | **53.87&plusmn;0.17** | **53.35&plusmn;0.24** | **50.88&plusmn;0.27** | **51.41&plusmn;0.37** | **57.77&plusmn;0.22** | **51.3&plusmn;0.26** |
>
> Table S15: Average robust accuracy (%) and standard deviation on CIFAR-10 dataset with WRN-32-10.
>
> | Method      |          NAT          |   $\operatorname{PGD}^{20}$     |   $\operatorname{PGD}^{100}$    |          MIM          |          CW          |   $\operatorname{APGD}_{CE}$    |  $\operatorname{APGD}_{DLR}$   |   $\operatorname{APGD}_{T}$    |    $\operatorname{FAB}_{T}$    |        Square         |          AA          |
> | :---------- | :-------------------: | :-------------------: | :-------------------: | :-------------------: | :------------------: | :-------------------: | :-------------------: | :-------------------: | :-------------------: | :-------------------: | :------------------: |
> | AT                  | **87.32&plusmn;0.21** |  49.01&plusmn;0.33   |  48.83&plusmn;0.27   |  48.25&plusmn;0.17   |   52.8&plusmn;0.25    |   54.17&plusmn;0.26   |   53.09&plusmn;0.36   |   48.34&plusmn;0.33   |   49.00&plusmn;0.43   |   57.5&plusmn;0.18    |   48.17&plusmn;0.48   |
> | SEAT (deteriorated) |   85.28&plusmn;0.42   |  55.68&plusmn;0.42   |  55.57&plusmn;0.19   |   55.6&plusmn;0.23   |   53.01&plusmn;0.41   |   54.12&plusmn;0.54   |   53.54&plusmn;0.28   |   49.95&plusmn;0.67   |   50.02&plusmn;0.75   |   57.81&plusmn;0.33   |   49.96&plusmn;0.31   |
> | **SEAT**            |   86.44&plusmn;0.12   | **59.84&plusmn;0.2** | **59.8&plusmn;0.16** | **60.87&plusmn;0.1** | **58.95&plusmn;0.34** | **57.57&plusmn;0.18** | **57.74&plusmn;0.29** | **55.06&plusmn;0.27** | **55.53&plusmn;0.36** | **62.26&plusmn;0.23** | **55.67&plusmn;0.22** |

---

> > ### Author Response · Authors · 2021-11-19
> > **Response to reviewer E4Bu (2/4)**
> >
> > Q2: More details for experimental settings, e.g., whether the author reported best or the last epoch results? What loss function was used for SEAT? And more ablation studies discussing different combinations (e.g., SEAT by TRADES, SEAT + CutMix by TRADES).
> >
> > A2: Thanks for the kind reminder. All the results reported in Tables 1 and 2 were from the best epochs and are averaged over 5 runs. We use cross-entropy to train our model. According to your suggestion, we report the results under different mixed strategies on CIFAR-10 dataset with ResNet18 in Table S16. Overall, SEAT+TRADES and SEAT+TRADES+CutMix do not perform well. We believe the reason is that TRADES cannot well adapt to learning strategies other than the staircase one for its history snapshots of TRADES trained under other strategies is much inferior to the ones trained under the staircase learning rate strategy. However, MART is in harmony with other strategies. The combination of SEAT, MART and CutMix achieves the best or the second-best robustness against almost all types of attacks.
> >
> > Table S16: Average robust accuracy (%) and standard deviation under different mixed strategies on CIFAR-10 dataset with ResNet18.
> >
> > | Method      |          NAT          |   $\operatorname{PGD}^{20}$     |   $\operatorname{PGD}^{100}$    |          MIM          |          CW          |   $\operatorname{APGD}_{CE}$    |  $\operatorname{APGD}_{DLR}$   |   $\operatorname{APGD}_{T}$    |    $\operatorname{FAB}_{T}$    |        Square         |          AA          |
> > | :---------- | :-------------------: | :-------------------: | :-------------------: | :-------------------: | :------------------: | :-------------------: | :-------------------: | :-------------------: | :-------------------: | :-------------------: | :------------------: |
> > | SEAT               | **83.7&plusmn;0.13** |  56.02&plusmn;0.11   |   55.97&plusmn;0.07   |   57.13&plusmn;0.12   | **54.38&plusmn;0.1** |   53.87&plusmn;0.17   |   53.35&plusmn;0.24   |  50.88&plusmn;0.27   |   51.41&plusmn;0.37   |   57.77&plusmn;0.22   |   51.3&plusmn;0.26   |
> > | SEAT+TRADES        |  81.21&plusmn;0.44   |  57.05&plusmn;0.28   |   57.0&plusmn;0.15    | **57.92&plusmn;0.12** |  52.75&plusmn;0.09   |   50.75&plusmn;0.25   |   50.36&plusmn;0.29   |  48.56&plusmn;0.37   |   49.45&plusmn;0.65   |   54.45&plusmn;0.58   |  49.91&plusmn;0.17   |
> > | SEAT+MART          |  78.94&plusmn;0.14   |  56.92&plusmn;0.46   |   56.88&plusmn;0.29   |   57.24&plusmn;0.48   |  52.09&plusmn;0.66   |   54.06&plusmn;0.56   |   53.77&plusmn;0.25   |  51.03&plusmn;0.44   |   51.11&plusmn;0.41   |   57.76&plusmn;0.08   |   51.7&plusmn;0.35   |
> > | SEAT+TRADES+CutMix |  78.22&plusmn;0.33   |  57.14&plusmn;0.21   |   57.09&plusmn;0.45   |   57.11&plusmn;0.39   |  53.17&plusmn;0.41   |   52.67&plusmn;0.82   |   51.86&plusmn;0.5    |  50.91&plusmn;0.17   |   50.23&plusmn;0.29   |   55.12&plusmn;0.46   |  50.01&plusmn;0.44   |
> > | SEAT+MART+CutMix   |  75.87&plusmn;0.24   | **57.3&plusmn;0.16** | **57.29&plusmn;0.43** |   57.47&plusmn;0.18   |   53.13&plusmn;0.2   | **54.33&plusmn;0.33** | **53.98&plusmn;0.61** | **51.2&plusmn;0.73** | **51.42&plusmn;0.17** | **58.01&plusmn;0.08** | **52.1&plusmn;0.22** |

---

> > > ### Author Response · Authors · 2021-11-19
> > > **Response to reviewer E4Bu (3/4)**
> > >
> > > Q3: Requirement of PoE involving with TRADES trained with different $\lambda$s.
> > >
> > > A3: Thank you for the constructive suggestion for us. It is a good idea to combine the outputs of TRADES with different $\lambda$​​. We have added the relevant experiments using both ResNet18 and WRN-32-10 on CIFAR10 in the modified paper (Tables 1 and 2). We also provide the results in Tables S17 and S18 for your convenience. Overall, results of PoE with different TRADES models show its great robustness against the CW and the most powerful AA attacks, while PoE with FAT and GAIRAT performs better than the TRADES one on the natural and the PGD testing.
> > >
> > > Table S17: Comparison of our algorithm with different defense methods using ReshaNet18 on CIFAR10. The maximum perturbation is $\varepsilon= 8/255$​​​​.  Average accuracy rates (in %) and standard deviations have shown that the proposed SEAT method greatly improves the robustness of the model.
> > >
> > > | Method                                |          NAT          |   $\operatorname{PGD}_{20}$    |   $\operatorname{PGD}_{100}$   |          MIM          |          CW          |          AA          |
> > > | ------------------------------------- | :-------------------: | :-------------------: | :-------------------: | :-------------------: | :------------------: | :------------------: |
> > > | AT                                    |   84.32&plusmn;0.23   |   48.29&plusmn;0.11   |   48.12&plusmn;0.13   |   47.95&plusmn;0.04   |  49.57&plusmn;0.15   |  44.37&plusmn;0.37   |
> > > | PoE (FAT+MART+TRADES+GAIRAT)          | **85.41&plusmn;0.29** |   55.2&plusmn;0.37    |   55.07&plusmn;0.24   |   54.33&plusmn;0.32   |  49.25&plusmn;0.16   |  46.17&plusmn;0.35   |
> > > | PoE (TRADES with different $\lambda$) |   83.57&plusmn;0.31   |   53.88&plusmn;0.45   |   53.82&plusmn;0.27   |   55.01&plusmn;0.18   |  52.72&plusmn;0.65   |   49.2&plusmn;0.24   |
> > > | **SEAT**                              |   83.7&plusmn;0.13    | **56.02&plusmn;0.11** | **55.97&plusmn;0.07** | **57.13&plusmn;0.12** | **54.38&plusmn;0.1** | **51.3&plusmn;0.26** |
> > >
> > > Table S18: Comparison of our algorithm with different defense methods using WRN-32-10 on CIFAR10. The maximum perturbation is $\varepsilon= 8/255$​​​.  Average accuracy rates (in %) and standard deviations have shown that SEAT also shows a great improvement on robustness.
> > >
> > > | Method                                 |          NAT          |    $\operatorname{PGD}_{20}$    |   $\operatorname{PGD}_{100}$   |         MIM          |          CW           |          AA           |
> > > | -------------------------------------- | :-------------------: | :------------------: | :------------------: | :------------------: | :-------------------: | :-------------------: |
> > > | AT                                     | **87.32&plusmn;0.21** |  49.01&plusmn;0.33   |  48.83&plusmn;0.27   |  48.25&plusmn;0.17   |   52.8&plusmn;0.25    |   48.17&plusmn;0.48   |
> > > | PoE (FAT+MART+TRADES+GAIRAT)           |   87.1&plusmn;0.25    |   55.75&plusmn;0.2   |  55.47&plusmn;0.19   |  56.04&plusmn;0.31   |   53.66&plusmn;0.18   |   49.44&plusmn;0.35   |
> > > | PoE (TRADES with different $\lambda$s) |   86.03&plusmn;0.37   |  54.26&plusmn;0.47   |  54.73&plusmn;0.21   |  55.01&plusmn;0.22   |   55.52&plusmn;0.18   |    53.2&plusmn;0.4    |
> > > | **SEAT**                               |   86.44&plusmn;0.12   | **59.84&plusmn;0.2** | **59.8&plusmn;0.16** | **60.87&plusmn;0.1** | **58.95&plusmn;0.34** | **55.67&plusmn;0.22** |
> > >
> > > Q4: In Table 1, I found at least TRADES is not sufficiently trained, its performance over NAT and AA should be at least better than 84\% and 49.5\% respectively.
> > >
> > > A4: Thanks for your careful reading. Actually, we trained all the baseline models without any tricks and the results in our paper are at least competitive with their original papers and some recent works. Take TRADES for example, the work [2] which offered comprehensive evaluations reported the accuracy rate of TRADES of 82.52%, 53.58% and 48.51% on natural, PGD<sup>10</sup> (easier than PGD<sup>20</sup>) and AA testing while ours reached the corresponding accuracy rate of 83.91%, 54.25% and 48.21%, respectively.

---

> > > > ### Author Response · Authors · 2021-11-19
> > > > **Response to reviewer E4Bu (4/4)**
> > > >
> > > > Q5: On Table 2, authors claim that "when compared with the results of ResNet18, CutMix performs much better when the model becomes larger." However, SEAT+CutMix didn't achieve the best against all the attacks (e.g., MIM), and we can also observe SEAT (86.44) performs better than SEAT+CutMix (84.81) over NAT, so CutMix's better performance over AA could be due to the trade-off between Natural accuracy and robustness. It is well possible that CutMix may benefit the model with more capacity, but this conclusion cannot be directly drawn based on the current observations on Table 2.
> > > >
> > > > A5: Actually, SEAT+CutMix on WRN-32-10 outperforms other methods *almost* all attack methods (except for MIM). Considering we evaluate various methods against fairly comprehensive attack methods, it is fair to say CutMix indeed boosts the robustness when cooperating with a large model. Moreover, similar to our experimental results, the work [1] also demonstrates that CutMix indeed benefits the model with more capacity though they performed the technique on the external data. This phenomenon springs from the fact that CutMix techniques tend to produce augmented views that are far away from the original image they augment, which means that they are a little hard for small neural networks to learn.
> > > >
> > > > Q6: Requirement of experiments on larger datasets, e.g., CIFAR-100.
> > > >
> > > > A6: Following your suggestion, we use ResNet18 to perform the experiment on CIFAR-100 with ResNet18 and the results shown in Table S19 demonstrate the effectiveness of SEAT for building a robust classifier.
> > > >
> > > > Table S19: Average robust accuracy (%) and standard deviation on CIFAR-100 dataset with ResNet18.
> > > >
> > > > | Method      |          NAT          |   $\operatorname{PGD}^{20}$     |   $\operatorname{PGD}^{100}$    |          MIM          |          CW          |   $\operatorname{APGD}_{CE}$    |  $\operatorname{APGD}_{DLR}$   |   $\operatorname{APGD}_{T}$    |    $\operatorname{FAB}_{T}$    |        Square         |          AA          |
> > > > | :---------- | :-------------------: | :-------------------: | :-------------------: | :-------------------: | :------------------: | :-------------------: | :-------------------: | :-------------------: | :-------------------: | :-------------------: | :------------------: |
> > > > | AT     | **60.1&plusmn;0.35** |   28.22&plusmn;0.3    |   28.27&plusmn;0.12   |   28.31&plusmn;0.41   |   24.87&plusmn;0.51   |   26.63&plusmn;0.29   |   24.13&plusmn;0.22   |   21.98&plusmn;0.3    |   23.87&plusmn;0.21   |   27.93&plusmn;0.12   |   23.91&plusmn;0.41   |
> > > > | TRADES |  59.93&plusmn;0.46   |   29.9&plusmn;0.41    |   29.88&plusmn;0.11   |   29.55&plusmn;0.25   |   26.14&plusmn;0.21   |   27.93&plusmn;0.44   |   25.43&plusmn;0.29   |   23.72&plusmn;0.45   |   25.16&plusmn;0.15   |   30.03&plusmn;0.32   |   24.72&plusmn;0.37   |
> > > > | MART   |  57.24&plusmn;0.64   |   30.62&plusmn;0.37   |   30.62&plusmn;0.17   |   30.83&plusmn;0.28   |   26.3&plusmn;0.29    |   29.91&plusmn;0.07   |   26.32&plusmn;0.24   |   24.28&plusmn;0.49   |   24.86&plusmn;0.66   |   28.28&plusmn;0.39   |   24.27&plusmn;0.21   |
> > > > | SEAT   |  56.28&plusmn;0.33   | **32.15&plusmn;0.17** | **32.12&plusmn;0.26** | **32.62&plusmn;0.15** | **29.68&plusmn;0.26** | **30.97&plusmn;0.18** | **29.62&plusmn;0.22** | **26.88&plusmn;0.23** | **27.71&plusmn;0.24** | **32.35&plusmn;0.34** | **27.87&plusmn;0.24** |
> > > >
> > > > Q7: Given the results reported for the proposed method show a great boost in robust performance, it would be nice to have the ability to view the training and evaluation code.
> > > >
> > > > A7: Thanks for your attention to our work. Since such an efficient method can bring ideal robustness and there still exists some potentials for the community to explore, we will release the code once the paper is accepted.
> > > >
> > > > [1] Rebuffi, Sylvestre-Alvise, et al. "Fixing data augmentation to improve adversarial robustness." arXiv preprint arXiv:2103.01946 (2021).
> > > >
> > > > [2] Pang, Tianyu, et al. "Bag of tricks for adversarial training." ICLR, 2021.

---

> ### Author Response · Authors · 2021-11-29
> **Need further clarification?**
>
> Thanks for your constructive comments. We have tried our best to address the concerns. Is there any unclear point that we should/could further clarify?

---

> > ### Comment · Reviewer_E4Bu · 2021-12-06
> > **Thank you for addressing concerns**
> >
> > Thanks a lot for providing thorough feedback to all my concerns! I appreciate your systematic experimental study on the adversarial robustness by using self-weight-ensembled method. As a result, I am willing to push my score up to a 6.

---

### Official Review · Reviewer_Ztw4 · 2021-11-02

**Correctness:** 3
**Technical Novelty And Significance:** 3
**Empirical Novelty And Significance:** 2
**Recommendation:** 6
**Confidence:** 3

**Main Review:**

The core algorithm SEAT proposed is to apply self-ensemble to model parameters during adversarial training. The paper demonstrates the effectiveness of the method with extensive experiments as well as with ablation studies.

For the main results in Section 4.2, it might be better to elaborate on how these numbers are obtained from experiments: e.g. are these all trained to the same step; are the numbers reported average over runs; what are the variances, etc, to give a better sense of how statistically significant the improvements are.

And comparing Table 1 and Table 2, it seems to be inconsistent around incorporating CutMix.

Since Section 3.3 tries to explain why the self-ensemble method could not be integrated into the routine working flow and argues the deterioration comes from the learning rate. It would also be interesting to know how it works with other learning rate schedules, such as with a warm-up stage at the beginning.

**Summary Of The Paper:**

This paper proposes the Self-Ensemble Adversarial Training (SEAT) which averages weights of history models for robustness. Compared to individual models or an ensemble of predictions from different classifiers, they argue that the proposed self-ensemble method provides a smoother loss landscape and better robustness. They also shed light on the deterioration of general self-ensemble models because of learning rates in the late phases.

**Summary Of The Review:**

The paper is well written with theoretical propositions and extensive experimental results.

---

> ### Author Response · Authors · 2021-11-19
> **Response to reviewer Ztw4 (1/2)**
>
> We’re very glad you had a positive initial impression and your suggestion is helpful to make our work more rigorous. We provide pointwise responses below.
>
> Q1: About how experimental results are obtained from experiments, such as the total training steps and the averaged results over multiple runs with their standard deviations.
>
> A1: Following your suggestion, we have added more details in the modified paper. We trained all the models to the same epoch and choose their best checkpoints to evaluate. All results in Tables 1 and 2 are computed with 5 individual trials. We omit the standard deviations of 5 runs as they are very small (<0.5%) for our SEAT method. We have added them to the modified paper to make the experiments more rigorous. And we also provide the results in Tables S11 and S12 for your convenience.
>
> Table S11: Comparison of our algorithm with different defense methods using ResNet18 on CIFAR10. Average accuracy rates (in %) and standard deviations have shown that the proposed SEAT method greatly improves the robustness of the model.
>
> | Method           |          NAT          |   $\operatorname{PGD}_{20}$    |   $\operatorname{PGD}_{100}$   |          MIM          |          CW          |          AA          |
> | ---------------- | :-------------------: | :-------------------: | :-------------------: | :-------------------: | :------------------: | :------------------: |
> | AT               |   84.32&plusmn;0.23   |   48.29&plusmn;0.11   |   48.12&plusmn;0.13   |   47.95&plusmn;0.04   |  49.57&plusmn;0.15   |  44.37&plusmn;0.37   |
> | TRADES           |   83.91&plusmn;0.33   |   54.25&plusmn;0.11   |   52.21&plusmn;0.09   |   55.65&plusmn;0.1    |  52.22&plusmn;0.05   |   48.2&plusmn;0.2    |
> | FAT              | **87.72&plusmn;0.14** |   46.69&plusmn;0.31   |   46.81&plusmn;0.3    |   47.03&plusmn;0.17   |  49.66&plusmn;0.38   |  43.14&plusmn;0.43   |
> | MART             |   83.12&plusmn;0.23   |   55.43&plusmn;0.16   |   53.46&plusmn;0.24   | **57.06&plusmn;0.2**  |  51.45&plusmn;0.29   |  48.13&plusmn;0.31   |
> | GAIRAT           |   83.4&plusmn;0.21    |   54.76&plusmn;0.42   |   54.81&plusmn;0.63   |   53.57&plusmn;0.31   |  38.71&plusmn;0.26   |  31.25&plusmn;0.44   |
> | PoE              |   85.41&plusmn;0.29   |   55.2&plusmn;0.37    |   55.07&plusmn;0.24   |   54.33&plusmn;0.32   |  49.25&plusmn;0.16   |  46.17&plusmn;0.35   |
> | CutMix (with WA) |   81.26&plusmn;0.44   |   52.77&plusmn;0.33   |   52.55&plusmn;0.25   |   53.01&plusmn;0.25   |  50.01&plusmn;0.55   |  47.38&plusmn;0.36   |
> | **SEAT**         |   83.7&plusmn;0.13    | **56.02&plusmn;0.11** | **55.97&plusmn;0.07** | **57.13&plusmn;0.12** | **54.38&plusmn;0.1** | **51.3&plusmn;0.26** |
> | **SEAT+CutMix**  |   81.53&plusmn;0.31   |   55.3&plusmn;0.27    |   54.82&plusmn;0.18   |   56.41&plusmn;0.17   |  53.83&plusmn;0.31   |   49.1&plusmn;0.44   |
>
> Table S12: Comparison of our algorithm with different defense methods using WRN-32-10 on CIFAR10. Average accuracy rates (in %) and standard deviations have shown that SEAT also shows a great improvement on robustness.
>
> | Method           |          NAT          |   $\operatorname{PGD}_{20}$   |   $\operatorname{PGD}_{100}$   |         MIM          |          CW           |          AA           |
> | ---------------- | :-------------------: | :------------------: | :-------------------: | :------------------: | :-------------------: | :-------------------: |
> | AT               |   87.32&plusmn;0.21   |  49.01&plusmn;0.33   |   48.83&plusmn;0.27   |  48.25&plusmn;0.17   |   52.8&plusmn;0.25    |   48.17&plusmn;0.48   |
> | TRADES           |   85.11&plusmn;0.77   |  54.58&plusmn;0.49   |   54.82&plusmn;0.38   |  55.67&plusmn;0.31   |   54.91&plusmn;0.21   |   52.19&plusmn;0.44   |
> | FAT              | **89.65&plusmn;0.04** |  48.74&plusmn;0.23   |   48.69&plusmn;0.18   |  48.24&plusmn;0.16   |   52.11&plusmn;0.71   |    46.7&plusmn;0.4    |
> | MART             |   84.26&plusmn;0.28   |  54.11&plusmn;0.58   |   54.13&plusmn;0.3    |   55.2&plusmn;0.22   |   53.41&plusmn;0.17   |   50.2&plusmn;0.36    |
> | GAIRAT           |   85.92&plusmn;0.69   |  58.51&plusmn;0.42   |   58.48&plusmn;0.34   |  58.37&plusmn;0.27   |   44.31&plusmn;0.22   |   39.64&plusmn;1.01   |
> | PoE              |   87.1&plusmn;0.25    |   55.75&plusmn;0.2   |   55.47&plusmn;0.19   |  56.04&plusmn;0.31   |   53.66&plusmn;0.18   |   49.44&plusmn;0.35   |
> | CutMix (with WA) |   82.79&plusmn;0.44   |  58.43&plusmn;1.21   |   58.2&plusmn;0.83    |  58.95&plusmn;0.57   |   58.32&plusmn;0.43   |   54.1&plusmn;0.82    |
> | **SEAT**         |   86.44&plusmn;0.12   |   59.84&plusmn;0.2   |   59.8&plusmn;0.16    | **60.87&plusmn;0.1** |   58.95&plusmn;0.34   |   55.67&plusmn;0.22   |
> | **SEAT+CutMix**  |   84.81&plusmn;0.18   | **60.2&plusmn;0.16** | **60.31&plusmn;0.12** |  60.53&plusmn;0.21   | **59.46&plusmn;0.24** | **56.03&plusmn;0.36** |

---

> > ### Author Response · Authors · 2021-11-19
> > **Response to reviewer Ztw4 (2/2)**
> >
> > Q2: Comparing Table 1 and Table 2, it seems to be inconsistent around incorporating CutMix.
> >
> > A2: Similar to our experimental results, as we observe experimental results from both [1] and ours, CutMix may benefit the model with more capacity. This phenomenon springs from the fact that CutMix techniques tend to produce augmented views that are far away from the original image they augment, which means that they are a little hard for small neural networks to learn.
> >
> >
> > Q3: How the SEAT method works with other learning rate schedules.
> >
> > A3: Thanks for your suggestion. Apart from showing results of the staircase, cosine and linear learning rate schedule in Figure 3 (a) in the original paper, we now provide two new strategies (1) the warming up learning rate schedule (Warmup) (2) the cyclic learning rate schedule (Cyclic) [2] and report all of them in Table S13. The effect of the warming up learning rate is marginal. When compared with the staircase one, the warmup strategy cannot generate diverse models in the later stages so the homogenization of the candidate models we mentioned in Section 3.2 cannot be fixed by the warmup strategy. On the contrary, those methods like cosine / linear / cyclic that provide relatively diverse models in the later stages can mitigate the issue, accounting for more robust ensemble models.
> >
> > Table S13: Average robust accuracy (%) under different learning strategies on CIFAR-10 dataset with ResNet18.
> >
> > | Method      |          NAT          |   $\operatorname{PGD}^{20}$     |   $\operatorname{PGD}^{100}$    |          MIM          |          CW          |   $\operatorname{APGD}_{CE}$    |  $\operatorname{APGD}_{DLR}$   |   $\operatorname{APGD}_{T}$    |    $\operatorname{FAB}_{T}$    |        Square         |          AA          |
> > | :---------- | :-------------------: | :-------------------: | :-------------------: | :-------------------: | :------------------: | :-------------------: | :-------------------: | :-------------------: | :-------------------: | :-------------------: | :------------------: |
> > | SEAT (Staircase) |  80.91   |      54.58       |       54.56       |   54.47   |   49.71   |       52.39       |       48.01        |      45.83       |      45.11      |   53.64   |   45.85   |
> > | SEAT (Cosine)    |   83.0   |      55.09       |       55.16       |   56.39   |   53.43   |       52.34       |        52.1        |      49.51       |      50.15      |   56.48   |   50.48   |
> > | SEAT (Linear)    | **83.7** |    **56.02**     |     **55.97**     | **57.13** | **54.38** |     **53.87**     |     **53.35**      |    **50.88**     |    **51.41**    | **57.77** | **51.3**  |
> > | SEAT (Warmup)    |  82.74   |      55.31       |       55.35       |   56.39   |   53.26   |       53.55       |       48.94        |      45.89       |      46.6       |   54.94   |   45.82   |
> > | SEAT (Cyclic)    |  83.14   |    **56.03**     |     **55.79**     | **56.99** | **54.01** |     **53.72**     |      **53.1**      |    **50.66**     |    **51.02**    | **57.75** | **51.44** |
> >
> > [1] Rebuffi, Sylvestre-Alvise, et al. "Fixing data augmentation to improve adversarial robustness." arXiv preprint arXiv:2103.01946 (2021).
> >
> > [2] Rice, Leslie, Eric Wong, and Zico Kolter. "Overfitting in adversarially robust deep learning." ICML, 2020.

---

> ### Author Response · Authors · 2021-11-29
> **Need further clarification?**
>
> Thanks for your constructive comments. We have tried our best to address the concerns. Is there any unclear point that we should/could further clarify?

---

### Official Review · Reviewer_K6Do · 2021-11-02

**Correctness:** 3
**Technical Novelty And Significance:** 4
**Empirical Novelty And Significance:** 2
**Recommendation:** 5
**Confidence:** 4

**Main Review:**

Please find my review of the paper *"Self-Ensemble Adversarial Training for Improved Robustness"* below. Unfortunately, I do not find the current version of the manuscript to be ready for publication at ICLR. As outlined below, the main contributor to this decision are the clarifications required to understand the proposed method, the formality of the theory sections, and the somewhat lacking experiment section.

In addition to these three points, lack of clarity in exposition also played an important role in my final decision. The current version of the manuscript contains a number of overly verbose or oddly formalised sentences that make the paper hard to follow. In the section "Small Notes" below, I've listed a number of these issues, but this is certainly not an exhaustive list.

### Proposed method is hard to understand

As I also said in the summary, if I understand correctly, the authors proposed method for improving the adversarial robustness of a model is to keep a running average of the models weights throughout training. Based on the initial discussion in the abstract and introduction this is very far from the ensembling approach that I was expecting.

Based on the presented algorithm box, I would argue that the main contribution of the paper is showing that using the SGD optimiser augmented with a momentum term improves the adversarial robustness of a model compared to when only plain SGD is used. The smoothness argument discussed in the paper also supports the view of the contribution being an augmented optimiser.

### Formality theoretical contributions

The paper proposes a number of theorems and propositions that I personally found to be written in a slightly weird form. This made it specifically hard to judge the correctness and usefulness of these propose theorems.

A clear example of this is found in Theorem 1, which ends with: *"SEAT will hardly be approximate to the average prediction of history networks".* In this case it is unclear to me what is meant with *hardly be approximate.*

In general this ties in with my earlier comment concerning the clarity of the exposition.

### Experimental results are lacking

While the authors did an admirable job by providing a exhaustive list of baseline methods for improving the adversarial robustness of a classifier, I believe the experimental evaluation still to be somewhat lacking. Specifically, as suggested in numerous papers discussing the most effective way of evaluating the effectiveness of a new defence strategy, the authors unfortunately do not include an adaptive attack in their evaluation. Specifically, I believe it to be important to study adaptive attacks that would be able to deal with the smoothness of the loss landscape.

In addition to this, I also believe that the experimental evaluation would benefit from doing multiple runs and reporting the standard deviation on the accuracy of these runs. This is especially important considering that the paper is not accompanied by code.

### Additional notes:

- What is meant by extremely imperceptible?
- Depicting adversarial training as "memorising" adversarial examples seems a very rough over simplification.
- First line of second paragraph of introduction is not grammatically correct.
- Sec 3.2, what is meant by: *"However, such a simple moving average cannot keenly capture the latest change, lagging behind the latest states by half the sample width."?*
- What is meant by this part of proposition 1:  "is ”almost” at least of the first order of smallness."?
- Remark 1: If the later settings are always more robust, why would we need the earlier models?
- Paragraph after remark 1, "uesd" → "used"

# Update after rebuttal.
The theoretical contributions of this paper are still not clear to me. However, based on the discussion with the authors and other reviewers, I will increase my score.

**Summary Of The Paper:**

The authors of the paper propose a new method for improving the robustness of a model against adversarial attacks. If I understood correctly, the proposed method is a combination of both adversarial training and model ensembling, where in this instance the ensembling is performed by maintaining a moving average of the weights of the model at past time steps.

In addition to some providing some theorems concerning their proposed method, the authors empirically show that the trained SEAT models are competitive with current methods for improving the adversarial robustness of the method.

**Summary Of The Review:**

The paper lacks clarity which makes it specifically hard to judge the contributions of the paper. In addition to this, the paper experimental evaluations requires more work.

---

> ### Author Response · Authors · 2021-11-19
> **Response to reviewer K6Do (1/4)**
>
> Thanks for your careful reading. We provide pointwise responses below. You might misunderstand the core idea of our algorithm so we make some clarifications to help you understand our work better.
>
> Q1: Based on the presented algorithm box, the main contribution of the paper is using the SGD optimiser augmented with a momentum term improves the adversarial robustness of a model compared to when only plain SGD is used.
>
> A1: I am afraid you may **misunderstood** the main idea of our algorithm. Our new ensemble method averages *parameters* of all history model snapshots through the training process to build an ensemble model on the fly, *rather than* averages *outputs* of multiple fully trained models to make the final prediction as the traditional ensemble methods do. That significantly reduces the time spent on training multiple networks and the memory to store the results of models. Moreover, $\tilde{\boldsymbol\theta}$​ in the algorithm box will *not* be assigned to the model at the following step for optimization (e.g. $\boldsymbol\theta_{t+1}:=\tilde{\boldsymbol\theta}$​). We just utilize $\boldsymbol\theta_{1,2,\cdots,t}$​ to update $\tilde{\boldsymbol\theta}$​ and then set $\tilde{\boldsymbol\theta}$​ aside through the training process, which is totally different from using a new optimizer with a momentum term like Adam. In summary, the averaged output methods are much slower and occupy storage space when compared with our online version. Besides, our experimental results also demonstrate that the output-averaged method underperforms our parameter-averaged method. It may be a promising way to rethink the effectiveness of the SEAT method from the perspective of optimization because all history snapshots, in other words, are local minima, but they do not conflict with each other.
>
> Q2: A number of theorems and propositions in a slightly weird form. For example, the meaning of "hardly be approximate" in Theorem 1 is unclear.
>
> A2: Basically, our theorem expounds on the correlation between the averaged prediction and the prediction of the parameter-averaged model. We find that the difference of these two can be of the second order of smallness in some special cases ($\sum_{t=1}^{T} (\beta_{t}\boldsymbol\xi^{\mathrm{T}})=\boldsymbol 0$​​). In the self-ensemble setting, the averaged output will be affected by the homogenization of later models, resulting in poor performance. Therefore, as long as these cases are excluded, the output of averaged parameters will not be as bad as the average output. The SEAT method is more difficult to achieve these extreme cases (if and only if $\beta_{i}=(1-\alpha) \alpha^{i-1}$​​ for $i \in\\{1,2,\cdots,T\\}$ ​) than the ordinary averaged parameter ensemble method. We have revised the expression of some theorems and shown the extreme cases to make statements much clearer.

---

> > ### Author Response · Authors · 2021-11-19
> > **Response to reviewer K6Do (2/4)**
> >
> > Q3: The authors do not include an adaptive attack in their evaluation and lack in experiments.
> >
> > A3: Actually, we performed AutoAttack (AA) in the original paper which is an ensemble of four diverse attacks and includes two adaptive methods to reliably evaluate robustness. We also perform each component of AA on CIFAR-10 dataset with both ResNet18 and WRN-32-10, including three parameter-free versions of PGD with the CE, DLR, targeted-CE loss with 9 target classes loss (APGD_CE, APGD_DLR, APGD_T), the targeted version of FAB (FAB_T) and an existing complementary Square [3]. Results are shown in the following Table S6 and S7. And our SEAT outperforms other methods against all components of AA. We also use ResNet18 to perform the experiment on CIFAR-100 and the results shown in Table S8 demonstrate the effectiveness of SEAT for building a robust classifier.
> >
> > Table S6: Average robust accuracy (%) and standard deviation against each component of AA on CIFAR-10 dataset with ResNet18.
> >
> > | Method   |   $\operatorname{APGD}_{CE}$    |  $\operatorname{APGD}_{DLR}$   |   $\operatorname{APGD}_{T}$    |    $\operatorname{FAB}_{T}$    |        Square         |          AA          |
> > | :------- | :-------------------: | :-------------------: | :-------------------: | :-------------------: | :-------------------: | :------------------: |
> > | AT       |   47.47&plusmn;0.35   |   48.57&plusmn;0.18   |   45.14&plusmn;0.31   |   46.17&plusmn;0.11   |   54.21&plusmn;0.15   |  44.37&plusmn;0.37   |
> > | TRADES   |   53.47&plusmn;0.21   |   50.89&plusmn;0.26   |   47.93&plusmn;0.36   |   48.53&plusmn;0.43   |   55.75&plusmn;0.21   |   48.2&plusmn;0.2    |
> > | MART     |   52.98&plusmn;0.13   |   50.36&plusmn;0.3    |   48.17&plusmn;0.72   |   49.39&plusmn;0.28   |   55.73&plusmn;0.51   |  48.13&plusmn;0.31   |
> > | **SEAT** | **53.87&plusmn;0.17** | **53.35&plusmn;0.24** | **50.88&plusmn;0.27** | **51.41&plusmn;0.37** | **57.77&plusmn;0.22** | **51.3&plusmn;0.26** |
> >
> > Table S7: Average robust accuracy (%) and standard deviation against each component of AA on CIFAR-10 dataset with WRN-32-10.
> >
> > | Method   |   $\operatorname{APGD}_{CE}$    |  $\operatorname{APGD}_{DLR}$   |   $\operatorname{APGD}_{T}$    |    $\operatorname{FAB}_{T}$    |        Square         |          AA          |
> > | :------- | :-------------------: | :-------------------: | :-------------------: | :-------------------: | :-------------------: | :-------------------: |
> > | AT       |   49.17&plusmn;0.26   |   50.09&plusmn;0.36   |   47.34&plusmn;0.33   |   48.00&plusmn;0.43   |   56.5&plusmn;0.18    |   48.17&plusmn;0.48   |
> > | TRADES   |   55.38&plusmn;0.43   |   55.55&plusmn;0.42   |   52.2&plusmn;0.13    |   53.11&plusmn;0.72   |   59.47&plusmn;0.17   |   52.19&plusmn;0.44   |
> > | MART     |   55.2&plusmn;0.32    |   55.41&plusmn;0.4    |   51.99&plusmn;0.3    |   52.88&plusmn;0.63   |   59.01&plusmn;0.38   |   50.2&plusmn;0.36    |
> > | **SEAT** | **57.57&plusmn;0.18** | **57.74&plusmn;0.29** | **55.06&plusmn;0.27** | **55.53&plusmn;0.36** | **62.26&plusmn;0.23** | **55.67&plusmn;0.22** |
> >
> > Table S8: Average robust accuracy (%) and standard deviation on CIFAR-100 dataset with ResNet18.
> >
> > | Method |         NAT          |   $\operatorname{PGD}_{20}$    |   $\operatorname{PGD}_{100}$   |          MIM          |          CW           | $\operatorname{APGD}_{CE}$    |  $\operatorname{APGD}_{DLR}$   |   $\operatorname{APGD}_{T}$    |    $\operatorname{FAB}_{T}$    |        Square         |          AA          |
> > | :----- | :------------------: | :-------------------: | :-------------------: | :-------------------: | :-------------------: | :-------------------: | :-------------------: | :-------------------: | :-------------------: | :-------------------: | :-------------------: |
> > | AT     | **60.1&plusmn;0.35** |   28.22&plusmn;0.3    |   28.27&plusmn;0.12   |   28.31&plusmn;0.41   |   24.87&plusmn;0.51   |   26.63&plusmn;0.29   |   24.13&plusmn;0.22   |   21.98&plusmn;0.3    |   23.87&plusmn;0.21   |   27.93&plusmn;0.12   |   23.91&plusmn;0.41   |
> > | TRADES |  59.93&plusmn;0.46   |   29.9&plusmn;0.41    |   29.88&plusmn;0.11   |   29.55&plusmn;0.25   |   26.14&plusmn;0.21   |   27.93&plusmn;0.44   |   25.43&plusmn;0.29   |   23.72&plusmn;0.45   |   25.16&plusmn;0.15   |   30.03&plusmn;0.32   |   24.72&plusmn;0.37   |
> > | MART   |  57.24&plusmn;0.64   |   30.62&plusmn;0.37   |   30.62&plusmn;0.17   |   30.83&plusmn;0.28   |   26.3&plusmn;0.29    |   29.91&plusmn;0.07   |   26.32&plusmn;0.24   |   24.28&plusmn;0.49   |   24.86&plusmn;0.66   |   28.28&plusmn;0.39   |   24.27&plusmn;0.21   |
> > | SEAT   |  56.28&plusmn;0.33   | **32.15&plusmn;0.17** | **32.12&plusmn;0.26** | **32.62&plusmn;0.15** | **29.68&plusmn;0.26** | **30.97&plusmn;0.18** | **29.62&plusmn;0.22** | **26.88&plusmn;0.23** | **27.71&plusmn;0.24** | **32.35&plusmn;0.34** | **27.87&plusmn;0.24** |

---

> > > ### Author Response · Authors · 2021-11-19
> > > **Response to reviewer K6Do (3/4)**
> > >
> > > Q4: Requirement of doing multiple runs and reporting the standard deviation on the accuracy of these runs.
> > >
> > > A4: Sorry for the confusion. All results in Tables 1 and 2 are computed with 5 individual trials. We omit the standard deviations of 5 runs as they are very small (<0.5%) for our SEAT method. We have added them in the modified paper and also provided the results in Tables S9 and S10 for your convenience.
> > >
> > > Table S9: Comparison of our algorithm with different defense methods using ResNet18 on CIFAR10. The maximum perturbation is $\varepsilon= 8/255$​​​.  Average accuracy rates (in %) and standard deviations have shown that the proposed SEAT method greatly improves the robustness of the model.
> > >
> > > | Method           |          NAT          |   $\operatorname{PGD}_{20}$    |   $\operatorname{PGD}_{100}$   |          MIM          |          CW          |          AA          |
> > > | ---------------- | :-------------------: | :-------------------: | :-------------------: | :-------------------: | :------------------: | :------------------: |
> > > | AT               |   84.32&plusmn;0.23   |   48.29&plusmn;0.11   |   48.12&plusmn;0.13   |   47.95&plusmn;0.04   |  49.57&plusmn;0.15   |  44.37&plusmn;0.37   |
> > > | TRADES           |   83.91&plusmn;0.33   |   54.25&plusmn;0.11   |   52.21&plusmn;0.09   |   55.65&plusmn;0.1    |  52.22&plusmn;0.05   |   48.2&plusmn;0.2    |
> > > | FAT              | **87.72&plusmn;0.14** |   46.69&plusmn;0.31   |   46.81&plusmn;0.3    |   47.03&plusmn;0.17   |  49.66&plusmn;0.38   |  43.14&plusmn;0.43   |
> > > | MART             |   83.12&plusmn;0.23   |   55.43&plusmn;0.16   |   53.46&plusmn;0.24   | **57.06&plusmn;0.2**  |  51.45&plusmn;0.29   |  48.13&plusmn;0.31   |
> > > | GAIRAT           |   83.4&plusmn;0.21    |   54.76&plusmn;0.42   |   54.81&plusmn;0.63   |   53.57&plusmn;0.31   |  38.71&plusmn;0.26   |  31.25&plusmn;0.44   |
> > > | PoE              |   85.41&plusmn;0.29   |   55.2&plusmn;0.37    |   55.07&plusmn;0.24   |   54.33&plusmn;0.32   |  49.25&plusmn;0.16   |  46.17&plusmn;0.35   |
> > > | PoE (TRADES)     |   83.57&plusmn;0.31   |   53.88&plusmn;0.45   |   53.82&plusmn;0.27   |   55.01&plusmn;0.18   |  52.72&plusmn;0.65   |   49.2&plusmn;0.24   |
> > > | CutMix (with WA) |   81.26&plusmn;0.44   |   52.77&plusmn;0.33   |   52.55&plusmn;0.25   |   53.01&plusmn;0.25   |  50.01&plusmn;0.55   |  47.38&plusmn;0.36   |
> > > | **SEAT**         |   83.7&plusmn;0.13    | **56.02&plusmn;0.11** | **55.97&plusmn;0.07** | **57.13&plusmn;0.12** | **54.38&plusmn;0.1** | **51.3&plusmn;0.26** |
> > > | **SEAT+CutMix**  |   81.53&plusmn;0.31   |   55.3&plusmn;0.27    |   54.82&plusmn;0.18   |   56.41&plusmn;0.17   |  53.83&plusmn;0.31   |   49.1&plusmn;0.44   |
> > >
> > > Table S10: Comparison of our algorithm with different defense methods using WRN-32-10 on CIFAR10. The maximum perturbation is $\varepsilon= 8/255$​​​.  Average accuracy rates (in %) and standard deviations have shown that SEAT also shows a great improvement on robustness.
> > >
> > > | Method           |          NAT          |   $\operatorname{PGD}_{20}$   |   $\operatorname{PGD}_{100}$   |         MIM          |          CW           |          AA           |
> > > | ---------------- | :-------------------: | :------------------: | :-------------------: | :------------------: | :-------------------: | :-------------------: |
> > > | AT               |   87.32&plusmn;0.21   |  49.01&plusmn;0.33   |   48.83&plusmn;0.27   |  48.25&plusmn;0.17   |   52.8&plusmn;0.25    |   48.17&plusmn;0.48   |
> > > | TRADES           |   85.11&plusmn;0.77   |  54.58&plusmn;0.49   |   54.82&plusmn;0.38   |  55.67&plusmn;0.31   |   54.91&plusmn;0.21   |   52.19&plusmn;0.44   |
> > > | FAT              | **89.65&plusmn;0.04** |  48.74&plusmn;0.23   |   48.69&plusmn;0.18   |  48.24&plusmn;0.16   |   52.11&plusmn;0.71   |    46.7&plusmn;0.4    |
> > > | MART             |   84.26&plusmn;0.28   |  54.11&plusmn;0.58   |   54.13&plusmn;0.3    |   55.2&plusmn;0.22   |   53.41&plusmn;0.17   |   50.2&plusmn;0.36    |
> > > | GAIRAT           |   85.92&plusmn;0.69   |  58.51&plusmn;0.42   |   58.48&plusmn;0.34   |  58.37&plusmn;0.27   |   44.31&plusmn;0.22   |   39.64&plusmn;1.01   |
> > > | PoE              |   87.1&plusmn;0.25    |   55.75&plusmn;0.2   |   55.47&plusmn;0.19   |  56.04&plusmn;0.31   |   53.66&plusmn;0.18   |   49.44&plusmn;0.35   |
> > > | PoE (TRADES)     |   86.03&plusmn;0.37   |  54.26&plusmn;0.47   |   54.73&plusmn;0.21   |  55.01&plusmn;0.22   |   55.52&plusmn;0.18   |    53.2&plusmn;0.4    |
> > > | CutMix (with WA) |   82.79&plusmn;0.44   |  58.43&plusmn;1.21   |   58.2&plusmn;0.83    |  58.95&plusmn;0.57   |   58.32&plusmn;0.43   |   54.1&plusmn;0.82    |
> > > | **SEAT**         |   86.44&plusmn;0.12   |   59.84&plusmn;0.2   |   59.8&plusmn;0.16    | **60.87&plusmn;0.1** |   58.95&plusmn;0.34   |   55.67&plusmn;0.22   |
> > > | **SEAT+CutMix**  |   84.81&plusmn;0.18   | **60.2&plusmn;0.16** | **60.31&plusmn;0.12** |  60.53&plusmn;0.21   | **59.46&plusmn;0.24** | **56.03&plusmn;0.36** |

---

> > > > ### Author Response · Authors · 2021-11-19
> > > > **Response to reviewer K6Do (4/4)**
> > > >
> > > > Responses for additional notes:
> > > >
> > > > All the typos have been fixed in the modified paper. Thanks for the careful reading.
> > > >
> > > > Q5: What is meant by extremely imperceptible?
> > > >
> > > > A5: Here we just want to express the harmfulness of adversarial examples since the attack can be performed successfully under some extreme conditions (e.g. using only one pixel [1]), which is extremely imperceptible. To avoid the confusion, we revised the sentence in the modified paper.
> > > >
> > > > Q6: Depicting adversarial training as "memorising" adversarial examples seems a very rough over simplification.
> > > >
> > > > A6: Sorry for the confusion. We have modified the sentence in the manuscript.
> > > >
> > > > Q7: Sec 3.2, what is meant by: "However, such a simple moving average cannot keenly capture the latest change, lagging behind the latest states by half the sample width."?
> > > >
> > > > A7: One characteristic of the simple moving average is that it can eliminate the variation in the period if each component in the sequence has a periodic fluctuation [2]. But a perfectly regular cycle is rarely encountered, especially in our setting where SGD largely leads the classifier to find a better solution than the ones found by previous steps. That shows a trend that the newer weights are closer to the optimal while the older ones are farther away from the optimal. Thus, the components used are not centered around the mean. As for half the sample width, the oldest part and the newest part of participants in the ordered data stream divided by the mean account for 50% of the whole data stream respectively.
> > > >
> > > > Q8: What is meant by this part of proposition 1: "is ”almost” at least of the first order of smallness."?
> > > >
> > > > A8: As we proved in the supplementary material, the difference between the averaged output $\bar{f_{\mathcal{F}}}(x,y)$​ and the output of averaged weights $f_{\tilde{\boldsymbol\theta}}(x, y)$​ is of the first order of smallness except for some special cases. More specifically, the difference can achieve the second order of smallness only when $\sum_{t=1}^{T} (\beta_{t}\boldsymbol\xi^{\mathrm{T}})=\boldsymbol 0$​. And we also give the proof that the averaged prediction of multiple networks and the prediction of SEAT is of the second order of smallness if and only if $\beta_{i}=(1-\alpha) \alpha^{i-1}$​ for $i\in\\{1,2,\cdots,T\\}$​. To avoid confusion, we have deleted some ambiguous words in the revised version.
> > > >
> > > >
> > > > Q9: Remark 1: If the later settings are always more robust, why would we need the earlier models?
> > > >
> > > > A9: If we pick a *single* classifier through the training process to defend against several adversarial attacks, it is obvious that the relatively later models will achieve higher robustness. However, things change in the *ensemble* setting. When being constructed by later models only, an ensemble of models lacks diversity and suffers from the homogenization issue we mentioned in Section 3.1, resulting in the deterioration shown in Section 3.3. It is important to ensemble a suite of equally strong and diverse models. Besides, the earlier models can still provide useful information for positioning the optimal because they are on the trajectory of optimization. Based on these three aspects, we introduce earlier models to cope with the homogenization issue but assign relatively small weights to them according to chronological order.
> > > >
> > > > [1] Su, Jiawei, Danilo Vasconcellos Vargas, and Kouichi Sakurai. "One pixel attack for fooling deep neural networks." *IEEE Transactions on Evolutionary Computation* 23.5 (2019): 828-841.
> > > >
> > > > [2] Bennett, Carl A., and N. L. Franklin. "Statistical analysis." *Chemistry and the Chemical Industry. New York, John Wiley and Sons, Inc* (1954).
> > > >
> > > > [3] Andriushchenko, Maksym, et al. "Square attack: a query-efficient black-box adversarial attack via random search." ECCV, 2020.

---

### Official Review · Reviewer_np74 · 2021-11-08

**Correctness:** 4
**Technical Novelty And Significance:** 3
**Empirical Novelty And Significance:** 3
**Recommendation:** 8
**Confidence:** 4

**Details Of Ethics Concerns:**

Nothing.

**Main Review:**

# Strength:

This work proposes a novel ensemble method. Instead of training several individuals, it reuses the historical states during training, which saves computation costs. The method is novel and meaningful. It also performs theoretical analysis, which compares the difference between this new self-ensemble method and existing ensemble methods. The analysis and proof are easy to understand and meaningful when following the authors' logic. Experiments are performed with various network structures under different adversarial attacks.

# Weakness:

1. The exponential moving average (EMA) is used in this paper, which is also widely used to improve the test performance of models. Could the authors explain what improvements do they make on EMA?
2. The experiments include results under AutoAttack (AA), but there are only overall results reported. Could the authors also report the results of each component of AA?
3. The authors claim the method is efficient. Could they report the FLOPs or the throughput of the model?

=========

# Post rebuttal responses:

Thanks for the response. My concerns are well addressed. Thus, I raise my score to 8. The added experiments compared to vanilla EMA further demonstrate the effectiveness and novelty of the proposed SEAT. Additional experiments on each component of AA (one type of adaptive attack) present the reliability of the robustness. The FLOPs and running time convince me the efficiency of SEAT. Overall, I think the method is novel and meaningful. The theoretical analysis and proof are easy to understand and thoughtful.

**Summary Of The Paper:**

This work proposes Self-Ensemble Adversarial Training (SEAT), which utilizes the states of history models during training to efficiently improve the adversarial robustness. The reusing of historical states in the self-ensemble can save computation costs in contrast to previous ensemble methods that usually train several individuals separately. Theoretical analysis is also performed to explain the new self-ensemble method.

**Summary Of The Review:**

The idea of ensemble historical states is novel. The theoretical analysis is meaningful. The experiments show this method has good potential.

---

> ### Author Response · Authors · 2021-11-19
> **Response to reviewer np74 (1/2)**
>
> Thanks for your positive feedback on the solidarity, experiments, and organization, and we also appreciate your suggestions on overcoming the weaknesses. We provide a pointwise response to the Cons below.
>
> Q1: EMA is also widely used to improve the test performance of models. Could the authors explain what improvements do they make on EMA?
>
> A1: Following your suggestion, we have additionally reported the natural accuracy and robust accuracy with both ResNet18 and WRN-32-10 used the vanilla EMA and our SEAT method on CIFAR-10, respectively. When we compare the results of the vanilla EMA and our proposed SEAT in Table S1 and S2, the robust accuracy cannot be significantly and reliably improved if we directly use the vanilla EMA through adversarial training. Therefore, we analyze the underlying reason based on Proposition 2 and perform the piecewise linear learning rate schedule instead of the staircase one.
>
> Table S1: Average robust accuracy (%) and standard deviation on CIFAR-10 dataset with ResNet18.
>
> | Method      |          NAT          |   $\operatorname{PGD}^{20}$     |   $\operatorname{PGD}^{100}$    |          MIM          |          CW          |   $\operatorname{APGD}_{CE}$    |  $\operatorname{APGD}_{DLR}$   |   $\operatorname{APGD}_{T}$    |    $\operatorname{FAB}_{T}$    |        Square         |          AA          |
> | :---------- | :-------------------: | :-------------------: | :-------------------: | :-------------------: | :------------------: | :-------------------: | :-------------------: | :-------------------: | :-------------------: | :-------------------: | :------------------: |
> | AT          | **84.32&plusmn;0.23** |   48.29&plusmn;0.11   |   48.12&plusmn;0.13   |   47.95&plusmn;0.04   |  49.57&plusmn;0.15   |   47.47&plusmn;0.35   |   48.57&plusmn;0.18   |   45.14&plusmn;0.31   |   46.17&plusmn;0.11   |   54.21&plusmn;0.25   |  44.37&plusmn;0.37   |
> | Vanilla EMA |   80.91&plusmn;0.38   |   54.58&plusmn;0.71   |   54.56&plusmn;0.29   |   54.47&plusmn;0.39   |  49.71&plusmn;0.41   |   52.39&plusmn;0.26   |   48.01&plusmn;0.18   |   45.83&plusmn;0.52   |   45.11&plusmn;0.23   |   53.64&plusmn;0.44   |  45.85&plusmn;0.19   |
> | **SEAT**    |   83.7&plusmn;0.13    | **56.02&plusmn;0.11** | **55.97&plusmn;0.07** | **57.13&plusmn;0.12** | **54.38&plusmn;0.1** | **53.87&plusmn;0.17** | **53.35&plusmn;0.24** | **50.88&plusmn;0.27** | **51.41&plusmn;0.37** | **57.77&plusmn;0.22** | **51.3&plusmn;0.26** |
>
> Table S2: Average robust accuracy (%) and standard deviation on CIFAR-10 dataset with WRN-32-10.
>
> | Method      |          NAT          |   $\operatorname{PGD}^{20}$     |   $\operatorname{PGD}^{100}$    |          MIM          |          CW          |   $\operatorname{APGD}_{CE}$    |  $\operatorname{APGD}_{DLR}$   |   $\operatorname{APGD}_{T}$    |    $\operatorname{FAB}_{T}$    |        Square         |          AA          |
> | :---------- | :-------------------: | :-------------------: | :-------------------: | :-------------------: | :------------------: | :-------------------: | :-------------------: | :-------------------: | :-------------------: | :-------------------: | :------------------: |
> | AT          | **87.32&plusmn;0.21** |  49.01&plusmn;0.33   |  48.83&plusmn;0.27   |  48.25&plusmn;0.17   |   52.8&plusmn;0.25    |   54.17&plusmn;0.26   |   53.09&plusmn;0.36   |   48.34&plusmn;0.33   |   49.00&plusmn;0.43   |   57.5&plusmn;0.18    |   48.17&plusmn;0.48   |
> | Vanilla EMA |   85.28&plusmn;0.42   |  55.68&plusmn;0.42   |  55.57&plusmn;0.19   |   55.6&plusmn;0.23   |   53.01&plusmn;0.41   |   54.12&plusmn;0.54   |   53.54&plusmn;0.28   |   49.95&plusmn;0.67   |   50.02&plusmn;0.75   |   57.81&plusmn;0.33   |   49.96&plusmn;0.31   |
> | **SEAT**    |   86.44&plusmn;0.12   | **59.84&plusmn;0.2** | **59.8&plusmn;0.16** | **60.87&plusmn;0.1** | **58.95&plusmn;0.34** | **57.57&plusmn;0.18** | **57.74&plusmn;0.29** | **55.06&plusmn;0.27** | **55.53&plusmn;0.36** | **62.26&plusmn;0.23** | **55.67&plusmn;0.22** |

---

> > ### Author Response · Authors · 2021-11-19
> > **Response to reviewer np74 (2/2)**
> >
> > Q2: Could the authors also report the results of each component of AA?
> >
> > A2: Thanks for your suggestion. we have performed each component of AA on CIFAR-10 dataset with both ResNet18 and WRN-32-10, including three parameter-free versions of PGD with the CE, DLR, targeted-CE loss with 9 target classes loss (APGD_CE, APGD_DLR, APGD_T), the targeted version of FAB (FAB_T) and an existing complementary Square [2]. Results are shown in the following Table S3 and S4. And it is obvious that our SEAT outperforms other methods against all components of AA.
> >
> > Table S3: Average robust accuracy (%) and standard deviation against each component of AA on CIFAR-10 dataset with ResNet18.
> >
> > | Method   |   $\operatorname{APGD}_{CE}$    |  $\operatorname{APGD}_{DLR}$   |   $\operatorname{APGD}_{T}$    |    $\operatorname{FAB}_{T}$    |        Square         |          AA          |
> > | :------- | :-------------------: | :-------------------: | :-------------------: | :-------------------: | :-------------------: | :------------------: |
> > | AT       |   47.47&plusmn;0.35   |   48.57&plusmn;0.18   |   45.14&plusmn;0.31   |   46.17&plusmn;0.11   |   54.21&plusmn;0.15   |  44.37&plusmn;0.37   |
> > | TRADES   |   53.47&plusmn;0.21   |   50.89&plusmn;0.26   |   47.93&plusmn;0.36   |   48.53&plusmn;0.43   |   55.75&plusmn;0.21   |   48.2&plusmn;0.2    |
> > | MART     |   52.98&plusmn;0.13   |   50.36&plusmn;0.3    |   48.17&plusmn;0.72   |   49.39&plusmn;0.28   |   55.73&plusmn;0.51   |  48.13&plusmn;0.31   |
> > | **SEAT** | **53.87&plusmn;0.17** | **53.35&plusmn;0.24** | **50.88&plusmn;0.27** | **51.41&plusmn;0.37** | **57.77&plusmn;0.22** | **51.3&plusmn;0.26** |
> >
> > Table S4: Average robust accuracy (%) and standard deviation against each component of AA on CIFAR-10 dataset with WRN-32-10.
> >
> > | Method   |   $\operatorname{APGD}_{CE}$    |  $\operatorname{APGD}_{DLR}$   |   $\operatorname{APGD}_{T}$    |    $\operatorname{FAB}_{T}$    |        $\operatorname{Square}$         |         $\operatorname{AA}$           |
> > | :------- | :-------------------: | :-------------------: | :-------------------: | :-------------------: | :-------------------: | :-------------------: |
> > | AT       |   49.17&plusmn;0.26   |   50.09&plusmn;0.36   |   47.34&plusmn;0.33   |   48.00&plusmn;0.43   |   56.5&plusmn;0.18    |   48.17&plusmn;0.48   |
> > | TRADES   |   55.38&plusmn;0.43   |   55.55&plusmn;0.42   |   52.2&plusmn;0.13    |   53.11&plusmn;0.72   |   59.47&plusmn;0.17   |   52.19&plusmn;0.44   |
> > | MART     |   55.2&plusmn;0.32    |   55.41&plusmn;0.4    |   51.99&plusmn;0.3    |   52.88&plusmn;0.63   |   59.01&plusmn;0.38   |   50.2&plusmn;0.36    |
> > | **SEAT** | **57.57&plusmn;0.18** | **57.74&plusmn;0.29** | **55.06&plusmn;0.27** | **55.53&plusmn;0.36** | **62.26&plusmn;0.23** | **55.67&plusmn;0.22** |
> >
> > Q3: Requirement of reporting the FLOPs or the throughput of the model.
> >
> > A3: Thanks for your advice. We use the number of Multiply-Accumulate operations (MACs) in Giga (G) to compute the theoretical amount of multiply-add operations in DNNs, roughly GMACs = 0.5 * GFLOPs. Besides, we also provide the actual running time. All results run on a single NVIDIA GeForce RTX 3090 GPU. As shown in Table S5, the SEAT method takes negligible MACs and training time when compared with standard adversarial training.
> >
> > Table S5: Evaluation of time complexity of SEAT. Here we use the number of Multiply-Accumulate operations  (MACs) in Giga (G) to measure the running time complexity. And we also compute the actual training time with or without the SEAT method using ResNet18 and WRN-32-10 on a single NVIDIA GeForce RTX 3090 GPU.
> >
> > | Method           | MACs (G) | Training Time (mins) |
> > | :--------------- | :------: | :------------------: |
> > | ResNet18 (AT)    |   0.56   |         272          |
> > | ResNet18 (SEAT)  |   0.59   |         273          |
> > | WRN-32-10 (AT)   |   6.67   |         1534         |
> > | WRN-32-10 (SEAT) |   6.81   |         1544         |
> >
> > [1] Croce, Francesco, and Matthias Hein. "Minimally distorted adversarial examples with a fast adaptive boundary attack." ICML, 2020.
> >
> > [2] Andriushchenko, Maksym, et al. "Square attack: a query-efficient black-box adversarial attack via random search." ECCV, 2020.

---

> ### Author Response · Authors · 2021-11-29
> **Need further clarification?**
>
> Thanks for your constructive comments. We have tried our best to address the concerns. Is there any unclear point that we should/could further clarify?

---

### Decision · Program_Chairs · 2022-01-20

**Decision:**

Accept (Poster)

**Comment:**

This paper proposes Self-Ensemble Adversarial Training (SEAT) for yielding a robust classifier by averaging weights of history models. The solution is different from an ensemble of predictions of different adversarially trained models. The authors also provided theoretical and empirical evidence that the proposed self-ensemble method yields a smoother loss landscape and better robustness than both individual models and an ensemble of predictions from different classifiers.

The paper receives a mixed rating of 8-6-6-5 (after private discussion; initially it was 8-6-5-3), and all reviewers actively engaged in discussion. From the three positive reviewers, it is in general consensus that this paper has a clear motivation, is easy to follow, and owns reasonable (not exceptional) novelty. The negative reviewer poses a number of concerns, citing the absence of adaptive attack evaluation, the unclear difference between vanilla EMA and SEAT, and the proof of Proposition 1. The authors provided detailed responses and the negative reviewer was partially convinced (not fully) after viewing other comments.

AC carefully reads all discussions and feels this fall into a borderline case. The authors did solid work and there is no fatal concern as AC can see. The majority sentiment is that this is a good paper, just not an exciting one. Hence, the current recommendation is a borderline acceptance.